# Maximizing Communication Efficiency for Large-scale Training via 0/1 Adam

**Yucheng Lu**[*]
Cornell University

**Conglong Li**
Microsoft

**Minjia Zhang**
Microsoft

**Christopher De Sa**
Cornell University

**Yuxiong He**
Microsoft

## Abstract

1-bit gradient compression and local steps are two representative techniques that enable drastic communication reduction in distributed SGD. Their benefits, however, remain an open question on Adam-based large model pre-training (e.g. BERT and GPT). In this paper, we demonstrate the non-linearity in Adam causes slow convergence even when 1-bit compression or local steps are individually applied. To alleviate this limitation, we propose **0/1 Adam** that linearizes each Adam step via approximating its optimizer states using their stale estimates and linear correlation. **0/1 Adam** performs an Adam-like step to preserve the adaptivity, while its linearity allows utilizing 1-bit compression and local steps simultaneously for wall-clock time speed up. We provide convergence guarantee for **0/1 Adam** on smooth non-convex objectives. On various large-scale benchmarks such as BERT-Base, BERT-Large, GPT-2 pre-training and ImageNet, we demonstrate on up to 128 GPUs that **0/1 Adam** is able to reduce up to 87% of data volume, 54% of communication rounds, and achieve up to $2\times$ higher training throughput and end-to-end training time reduction compared to the state-of-the-art baseline 1-bit Adam; while enjoying the same statistical convergence speed and end task model accuracy on GLUE dataset and ImageNet validation set.

## 1 Introduction

Over the past few years, we have witnessed outstanding performance of foundation models on many applications. However, these models, including BERT Devlin et al. (2018) and GPT Radford et al. (2019a); Brown et al. (2020), usually have hundreds of millions or even billions of parameters and require to be trained on massive GPUs. For example, the largest dense transformer model, 530B MT-NLG Smith et al. (2022), was trained over 4000 GPUs in more than a month. At this scale, the expensive communication overhead across computing processors and servers hinders the scalability (Alistarh et al., 2017).

*1-bit gradient compression* and *local steps* are two representative methods to mitigate the communication bottleneck. 1-bit compression drastically reduces the communication volume by quantizing each value in gradients with ultra-low bits (i.e., as low as 1 bit) Seide et al. (2014); Bernstein et al. (2018a); and local steps alternatively saves the bandwidth by periodically skipping communication rounds(Stich, 2018). While these techniques demonstrate tremendous success on distributed SGD, their benefits over large-scale Adam-based model training, such as for BERT and GPT pre-training, remains an open question (Kingma and Ba, 2014; Wang et al., 2019a). Comparing to SGD where the model parameters are linearly dependent on the gradients, the non-linearity in Adam updates (Kingma and Ba, 2014) limits the direct usage of compression or local steps. In particular, this non-linearity incurs two challenges: 1) when aggressively compressing the gradient such as with 1-bit quantizer, all the coordinate-wise effect learning rate will become the same value, so that Adam no longer enjoys adaptive and fast convergence; 2) to ensure all parallel workers reach consensus on the optimizer states, which is critical for convergence, the existence of non-linearity incurs the overhead of iteratively synchronizing the states when using local steps.

Tang et al. (2021) undertook the first investigation of fixing this non-linearity towards compression and proposed **1-bit Adam**. The algorithm follows a two-stage training paradigm: first run Adam

---

[*]This work was done during Yucheng Lu's internship at Microsoft. Contact: yl2967@cornell.edu.

with full-precision communication (*full-precision stage*[1]); and then switch to 1 bit when the variance becomes stable (*compression stage*). While this paradigm avoids compressing non-linear information with a one-time frozen variance, the experimental results from (Tang et al., 2021) indicate the full-precision stage still incurs non-trivial overhead. Furthermore, 1-bit Adam is restricted in the scope of gradient compression, and cannot be trivially adapted when other techniques are used, such as local steps. Besides, the empirical success of (Tang et al., 2021) was not substantiated on GPT-style models, for instance, 175B GPT-3 Brown et al. (2020), 530B MT-NLG Smith et al. (2022), etc.

In this paper, we address this gap by proposing **0/1 Adam**. **0/1 Adam** breaks the barrier of non-linearity from two aspects: first it adaptively freezes variance, so that given agreement on a stale variance state, the parallel workers only need to communicate momentum that *is* linearly dependent on the model update; This technique allows reducing the previous two-stage compression scheme to a unified single stage; 2) it leverages the insight that in adjacent Adam steps, the changes to optimizer states are generally bounded, so that with frozen variance, parallel workers can linearly approximate momentum and parameter updates locally without additional synchronization. This further pushes the limit of communication reduction towards its extreme, achieving the state-of-the-art speed up for large-scale model training. To summarize, our contributions are as follows:

- We propose **0/1 Adam**, which addresses the limitations of previously proposed 1-bit Adam when applying aggressive 1-bit quantization and local steps (Section 4).
- We provide convergence guarantee of **0/1 Adam** on smooth and non-convex objectives (Section 5).
- We conduct experiments on a wide range of large-scale model training tasks, including BERT-Base, BERT-Large, GPT-2 pre-training and ImageNet. We demonstrate on up to 128 GPUs that **0/1 Adam** is able to reduce up to 87% of data volume, 54% of communication rounds, and achieve up to $2\times$ higher throughput and training time reduction compared to the state-of-the-art 1-bit Adam without compromising end-to-end model accuracy (Section 6).
- The 0/1 Adam optimizer and corresponding experimental scripts (e.g. BERT pre-training and GLUE finetuning) have been open sourced in a deep learning optimization library called DeepSpeed[2].

## 2 RELATED WORK

**Communication-efficient training.** There has been various lines of research focusing on improving communication efficiency in large-scale training, such as using asynchrony (Niu et al., 2011; Lian et al., 2015; Xie et al., 2020), decentralization (Lian et al., 2017; Lu and De Sa, 2021), gradient quantization (Alistarh et al., 2017; Wen et al., 2017), gradient sparsification (Wangni et al., 2017; Wang et al., 2018a), local steps (Stich, 2018; Lin et al., 2018), etc. In this paper we study the aggressive 1-bit compression, which was first introduced in (Seide et al., 2014) to speed up speech model training, where an algorithm called 1-bit SGD is proposed. After that, Wen et al. (2017) proposes adding 0 as an additional numerical level and Liu et al. (2018) discusses the use of zero-th order oracle in 1-bit SGD. Chen et al. (2019a); Balles and Hennig (2018); Xu and Kamilov (2019) study the correlation and combination between 1-bit SGD and other techniques. Convergence analysis on 1-bit SGD is given in (Bernstein et al., 2018a; Karimireddy et al., 2019; Safaryan and Richtárik, 2021). Bernstein et al. (2018b); Sohn et al. (2019); Le Phong and Phuong (2020); Lyu (2021) investigate the robustness of 1-bit SGD. Among all the variants of 1-bit communication, the design with error feedback mechanism has shown to work best both empirically (Seide et al., 2014) and theoretically (Karimireddy et al., 2019). Other lines of research applies 1-bit communication to various scenarios such as federated learning (Jin et al., 2020; Yue et al., 2021), decentralized learning (Lu and De Sa, 2020; Koloskova et al., 2019), meta learning (Fan et al., 2021), etc. Perhaps the closest works to this paper are (Tang et al., 2021; Li et al., 2021a), which propose using two-stage training to enable 1-bit Adam and 1-bit Lamb, respectively. Different from those two work, 0/1 Adam addresses non-linearity challenges in adaptive optimizers by considering both extreme quantization and local steps. Furthermore, we also study how to apply extreme communication compression on GPT-style models, which to the best our knowledge is still under-explored.

**Adaptive learning rate optimizers.** One of the most popular adaptive optimizers is Adam, which was first introduced in (Kingma and Ba, 2014). It uses both first and second moment information of stochastic gradient to perform optimizer steps and has shown significant benefits on training deep

---

[1] In the original 1-bit Adam paper, this stage is referred to as warmup stage. We use a slightly different term to avoid confusion with learning rate warmup.

[2] https://github.com/microsoft/DeepSpeed

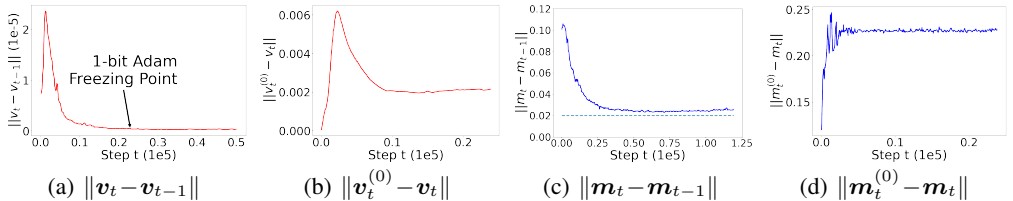

Figure 1: Momentum and variance Profiling for BERT-Large sequence 128 pretraining with original Adam using 64 GPUs. For variance, we profile two types of metrics: the first is the difference between local and global variance: $\|\boldsymbol{v}_t^{(0)} - \boldsymbol{v}_t\|$, where $\boldsymbol{v}_t^{(0)}$ and $\boldsymbol{v}_t$ denotes the variance term computed via local gradient on worker-0 and the gradient from full-precision AllReduce, respectively. We also profile the variance difference in adjacent step $\|\boldsymbol{v}_t - \boldsymbol{v}_{t-1}\|$. Similarly, we profile the same two metrics for the momentum.

learning models. Reddi et al. (2019) spots the issue of Adam convergence and provides a variant called AMSGrad while Zaheer et al. (2018) argues the Adam only converges with large batch sizes. Multiple lines of theoretical study on Adam are given in (Fang and Klabjan, 2019; Alacaoglu et al., 2020; Défossez et al., 2020). Additionally, Chen et al. (2018); Zhou et al. (2018a); Lu et al. (2020); Danilova et al. (2020); Zou et al. (2019) provide more general analysis on Adam-type optimizers. Subsequently, other variants of Adam are proposed in (Luo et al., 2019; Chen et al., 2019b; Huang et al., 2018; Wang et al., 2019b; Zhou et al., 2018b; Zhuang et al., 2021; 2020). Unlike these methods, which focus on improving the convergence of generic optimizations for DNN models, our work studies how to maximize the communication efficiency of Adam in large-scale distributed training settings.

## 3 A CLOSER LOOK AT NON-LINEARITY IN ADAM

In this section, we provide a more formal description on the problem setting and illustrate the limitations from the original Adam and the state-of-the-art 1-bit Adam (Tang et al., 2021).

**Problem Formulation.** In this paper, we consider the following optimization problem:

$$\min_{\boldsymbol{x} \in \mathbb{R}^d} f(\boldsymbol{x}) = \mathbb{E}_{\zeta \sim \mathcal{D}} f(\boldsymbol{x}; \zeta). \tag{1}$$

where $\boldsymbol{x}$ denotes the $d$-dimensional model. $\mathcal{D}$ denotes the training set and $f(\boldsymbol{x}; \zeta)$ is the loss incurred over sample $\zeta$ given model parameters $\boldsymbol{x}$. The structure of the problem naturally captures many of the model training problems.

**The non-linearity in Adam.** At step $t \geq 0$, denote $\boldsymbol{x}_t$ and $\boldsymbol{g}_t$ as the model parameters and stochastic gradient computed at step $t$, respectively. The update formula of SGD and Adam[3] can be summarized as:

$$\text{SGD update: } \boldsymbol{x}_{t+1} \leftarrow \boldsymbol{x}_t - \gamma \boldsymbol{g}_t. \tag{2}$$

$$\text{Adam update: } \boldsymbol{m}_{t+1} \leftarrow \beta_1 \boldsymbol{m}_t + (1 - \beta_1) \boldsymbol{g}_t,$$

$$\boldsymbol{v}_{t+1} \leftarrow \beta_2 \boldsymbol{v}_t + (1 - \beta_2)(\boldsymbol{g}_t)^2,$$

$$\boldsymbol{x}_{t+1} \leftarrow \boldsymbol{x}_t - \underbrace{\frac{\gamma}{\sqrt{\boldsymbol{v}_t + \epsilon}}}_{\text{effective learning rate}} \cdot \boldsymbol{m}_t, \tag{3}$$

where $\gamma$ is the learning rate, $\epsilon$ is a small constant to prevent zero division, $\beta_1$ and $\beta_2$ are tunable decaying factors. The linearity in SGD update implies when using compression or local steps, the potential noise from (accumulated) gradients is in the order of $O(\gamma)$, which approaches zero when learning rate is decaying or set to be small. By comparison, the two auxiliary optimizer states in Adam, momentum ($\boldsymbol{m}$) and variance ($\boldsymbol{v}$), introduce non-linearity in the model update.

Equation (3) gives the formula of Adam when running it sequentially. In a distributed setting with $n$ workers, $\boldsymbol{g}_t$ in Equation (3) is often computed in parallel on different workers. Mathematically, if we denote $\boldsymbol{g}_t^{(i)}$ as the stochastic gradient computed on the $i$-th worker at step $t$, then distributed Adam

---

[3]Note that in Adam, operations like division should act element-wise.

can be written as replacing $\boldsymbol{g}_t$ with $1/n\sum_{i=1}^{n}\boldsymbol{g}_t^{(i)}$ in Equation (3) as follows:

$$\boldsymbol{m}_{t+1} \leftarrow \beta_1\boldsymbol{m}_t + (1-\beta_1)\left(1/n\sum_{i=1}^{n}\boldsymbol{g}_t^{(i)}\right), \boldsymbol{v}_{t+1} \leftarrow \beta_2\boldsymbol{v}_t + (1-\beta_2)\left(1/n\sum_{i=1}^{n}\boldsymbol{g}_t^{(i)}\right)^2.$$

**Issue with non-linearity on 1-bit compression.** The main bottleneck in running distributed Adam is the accumulation of $1/n\sum_{i=1}^{n}\boldsymbol{g}_t^{(i)}$ since the gradients are usually high-dimensional. Based on the profiling results from (Tang et al., 2021; Li et al., 2021a), the communication of gradients could take up to 94% of the total training time on modern clusters. Gradient compression mitigates this issue by sending and averaging gradients with fewer bits. However, in Adam this causes the loss on the learning rate adaptivity. Consider using the aggressive 1-bit compression (Liu et al., 2018), which sends each gradient with only signs and a shared, usually the average over all the coordinates, magnitude. More specifically, denote $\mathcal{C}[\cdot]$ as the 1-bit compression, then

$$\mathcal{C}[\boldsymbol{a}] = \frac{\|\boldsymbol{a}\|_1}{d}\cdot\mathrm{sign}(\boldsymbol{a}), \forall \boldsymbol{a} \in \mathbb{R}^d. \tag{4}$$

It is straightforward to observe that naively applying 1 bit to compress gradients in the original Adam loses coordinate-wise adaptivity since sharing magnitude makes all the coordinates-wise learning rate $\gamma/\sqrt{\boldsymbol{v}_t + \epsilon}$ the same value. This makes Adam no difference than momentum SGD.

**Issue with non-linearity on local steps.** In SGD (Equation (2)), the model updates are linearly dependent on the gradients and has zero additional states. It implies with local steps, the parallel workers can entirely reach consensus after a single round of synchronization, even with compression (Basu et al., 2020). However, in Adam simply synchronizing the model can still leave the momentum and variance out-of-sync. This makes parallel workers fail to capture the global adaptivity when the system scales up. To give a more concrete example, we profile a full run of BERT-Large pre-training with original Adam, and summarize different metrics of momentum and variance in Figure 1. As shown in Figure 1(d) and 1(b), the difference between local and global optimizer states, momentum and variance, remain constants and do not decrease to zero.

**1-bit Adam and its limitations.** 1-bit Adam (Tang et al., 2021) is a state-of-the-art solution that addresses non-linearity in 1-bit compression. 1-bit Adam adopts a pre-conditioned variance state from running original Adam for $T_0$ steps first. The intuition there is that at later stage of training, the variance state becomes stable so that $\boldsymbol{v}_{T_0}$ can be a good approximation of variance state for the remaining steps. As paritally illustrated in Section 1, the full-precision stage of 1-bit Adam still presents non-trivial overhead. For instance: as illustrated in (Tang et al., 2021), when training BERT-Large on 64 GPUs using Ethernet, while the full-precision stage contains 15% of the total steps, it can take more than 50% of the entire training in terms of the wall-clock time[4]. Additionally, 1-bit Adam is restricted in the scope of compression, how it handles other techniques such as local steps remains open question.

## 4   0/1 ADAM

In this section, we give the full description of **0/1 Adam**. To maximize the communication efficiency, ideally we want an algorithm that enables adaptive convergence like Adam, while allowing aggressive compression (e.g. 1 bit) and requires no additional synchronization on the optimizer states when using local steps. **0/1 Adam** solves this problem from two aspects.

**Adaptive Variance Freezing.** To begin with, **0/1 Adam** creates a linear environment that freezes the variance adaptively. The intuition is leveraged from the observation in Figure 1(a): the change of variance over steps in Adam is generally smooth. While 1-bit Adam captures a reasonable variance estimate via one-time freezing, it is reasonable to also presume that before its freezing point, the variance within several adjacent steps will stay close due to its smoothness. This motivates us to extend the one-time freezing policy in 1-bit Adam into an adaptive one, by letting workers agree upon the freezing points from a given step index set $\mathcal{T}_{\boldsymbol{v}} \subseteq \{0, \cdots, T-1\}$. The frozen variance creates multiple intervals over training, during which the workers have agreement on the denominator (Equation (3)) and the only uncertainty is then left in the nominator that is linearly dependent on the model update, just like SGD.

---

[4]Concretely, it shows in (Tang et al., 2021) Section 7.1 that to train BERT-Large on 64 GPUs using Ethernet, the full-precision Adam takes 174.3 hours in total while 1-bit Adam takes 51.5 hours. By a simple calculation, we know that full-precision stage of 1-bit Adam takes approximately 26.37 hours while the compression stage takes 25.13 hours.

---

**Algorithm 1** Proposed **0/1 Adam** Algorithm

---

**Require:** local model on the $i$-th node $\boldsymbol{x}_0^{(i)}$, learning rate $\{\gamma_t\}_{t=1}^T$, $\boldsymbol{m}_0 = \boldsymbol{0}$, $\boldsymbol{v}_0 = \boldsymbol{0}$, auxiliary buffer $\boldsymbol{u}_0 = \boldsymbol{0}$, total number of iterations $T$, decaying factor $\beta_1, \beta_2$ from Adam, numerical constant $\epsilon$, variance update step index set $\mathcal{T}_{\boldsymbol{v}}$, synchronization step index set $\mathcal{T}_{\boldsymbol{u}}$, the most recent step with synchronization $t' = 0$.

1: **for** $t = 0, \cdots, T-1$ **do**
2:      Compute local stochastic gradient $\boldsymbol{g}_t^{(i)}$.
3:      Update momentum: $\boldsymbol{m}_{t+\frac{1}{2}}^{(i)} = \beta_1 \boldsymbol{m}_t^{(i)} + (1-\beta_1)\boldsymbol{g}_t^{(i)}$.
4:      Update model: $\boldsymbol{x}_{t+\frac{1}{2}}^{(i)} = \boldsymbol{x}_t^{(i)} - \gamma_t \boldsymbol{m}_t^{(i)} / \sqrt{\boldsymbol{v}_t + \epsilon}$.
5:      Update buffer: $\boldsymbol{u}_{t+\frac{1}{2}}^{(i)} = \boldsymbol{u}_t^{(i)} + \gamma_t \boldsymbol{m}_t^{(i)}$.
6:      **if** $t \in \mathcal{T}_{\boldsymbol{u}}$ **then**
7:          Perform 1-bit AllReduce: $\overline{\boldsymbol{u}}_{t+\frac{1}{2}} = \textbf{1bit-AllReduce}\left(\boldsymbol{u}_{t+\frac{1}{2}}^{(i)}\right)$.
8:          Approximate momentum with compressed buffer: $\boldsymbol{m}_{t+1}^{(i)} = \overline{\boldsymbol{u}}_{t+\frac{1}{2}} / \sum_{h=t'}^t \gamma_h$.
9:          Update model with compressed buffer: $\boldsymbol{x}_{t+1}^{(i)} = \boldsymbol{x}_{t'}^{(i)} - \overline{\boldsymbol{u}}_{t+\frac{1}{2}} / \sqrt{\boldsymbol{v}_t + \epsilon}$.
10:         Reset the auxiliary buffer: $\boldsymbol{u}_{t+1}^{(i)} = \boldsymbol{0}$.
11:         Update the synchronization step: $t' = t$.
12:      **else**
13:         $\boldsymbol{x}_{t+1}^{(i)} = \boldsymbol{x}_{t+\frac{1}{2}}^{(i)}; \boldsymbol{m}_{t+1}^{(i)} = \boldsymbol{m}_{t+\frac{1}{2}}^{(i)}; \boldsymbol{u}_{t+1}^{(i)} = \boldsymbol{u}_{t+\frac{1}{2}}^{(i)}$.
14:      **end if**
15:      **if** $t \in \mathcal{T}_{\boldsymbol{v}}$ **then**
16:          Perform full-precision AllReduce: $\overline{\boldsymbol{g}}_t = \textbf{AllReduce}\left(\boldsymbol{g}_t^{(i)}\right)$.
17:          Update the variance: $\boldsymbol{v}_{t+1} = \beta_2 \boldsymbol{v}_t + (1-\beta_2)(\overline{\boldsymbol{g}}_t)^2$.
18:      **else**
19:         Use the stale variance for the next iteration: $\boldsymbol{v}_{t+1} = \boldsymbol{v}_t$.
20:      **end if**
21: **end for**
22: **return** $\boldsymbol{x}_T$.

---

**Including 1-bit Compression and Local Steps.** With frozen variance, we make another observation based on Equation (3) that the model difference on workers will be linearly dependent to the momentum. So that, the momentum can be approximated locally rather than synchronized additionally based on the communicated model difference, given the premise that the change of momentum is not abrupt within close steps. Formally, denote $\boldsymbol{x}_t^{(i)}$, $\boldsymbol{m}_t^{(i)}$, $\boldsymbol{v}_t^{(i)}$ as the model, momentum, variance on worker $i$ at step $t$, respectively. Suppose all the workers are synchronized at step $t'$, then with frozen variance $\boldsymbol{v}$ over all the workers,

$$\boldsymbol{u}_t^{(i)} = \sum_{k=t'}^t \gamma_k \boldsymbol{m}_k^{(i)} \qquad \text{Actual sent tensors in the communication.}$$

$$\boldsymbol{x}_{t+1}^{(i)} = \boldsymbol{x}_{t'}^{(i)} - \frac{1/n \sum_{i=1}^n \boldsymbol{u}_t^{(i)}}{\sqrt{\boldsymbol{v} + \epsilon}} \qquad \text{Sync model parameters with the sent tensors.}$$

$$\boldsymbol{m}_{t+1}^{(i)} \approx \frac{1/n \sum_{i=1}^n \boldsymbol{u}_t^{(i)}}{\sum_{k=t'}^t \gamma_k} \qquad \text{Approximate momentum via linear estimates via sent tensor.}$$

where we omit the compression part for brevity. Combined with compression, we provide the full description of **0/1 Adam**[5] in Algorithm 1. Note that here we defer the details of 1-bit compression to Appendix A and treat it as a black-box procedure named **1bit-AllReduce** while the original full-precision AllReduce is referred to as **AllReduce**.

We also remark that although both techniques appear to be natural, to the best of our knowledge, we are the first to apply them to addressing the non-linearity challenges in 1-bit compression and local steps for maximizing the communication efficiency of Adam optimizer.

---

[5]The name comes from the fact that the algorithm can potentially reduce the per-parameter volume to some number between 0 and 1 bit on average.

# 5 CONVERGENCE ANALYSIS

In this section, we provide the convergence guarantee for **0/1 Adam** (Algorithm 1) under arbitrary freezing policy $\mathcal{T}_v$ and local steps policy $\mathcal{T}_u$. In the main paper, we provides the convergence rate in the general case. However, different $\mathcal{T}_v$ or $\mathcal{T}_u$ gives us the opportunity to obtain tighter bounds. We leave these discussion in the appendix. We start by making the following assumptions.

**Assumption 1.** *Lipschitzian gradient: $f(\cdot)$ is assumed to be with $L$-Lipschitzian gradients, which means* $\|\nabla f(\boldsymbol{x}) - \nabla f(\boldsymbol{y})\| \leq L\|\boldsymbol{x} - \boldsymbol{y}\|, \forall \boldsymbol{x}, \forall \boldsymbol{y}.$

**Assumption 2.** *Bounded variance: The stochastic gradient computed on each worker is unbiased and has bounded variance:$\mathbb{E}_{\zeta \sim \mathcal{D}}\|\nabla f(\boldsymbol{x}; \zeta) - \nabla f(\boldsymbol{x})\|^2 \leq \sigma^2, \quad \forall \boldsymbol{x}.$*

**Assumption 3.** *Bounded gradient: The infinity norm of stochastic gradient is bounded by a constant $G_\infty > 0$ such that $\|\boldsymbol{g}_t\|_\infty \leq G_\infty, \forall t.$*

**Assumption 4.** *Compression error in Algorithm 1: For arbitrary $\boldsymbol{x} \in \mathbb{R}^d$, there exists a constant $\Delta$, such that the output of compressor $\mathcal{C}[\cdot]$ has the following error bound: $\mathbb{E}\|\mathcal{C}[\boldsymbol{x}] - \boldsymbol{x}\|^2 \leq \Delta^2.$*

**Assumption 5.** *Given ordered set $\mathcal{T}_u$, denote $t_j$ as the $j$-th element in $\mathcal{T}_u$, we assume there exists a constant $H \geq 0$, it holds that $\max_{1 \leq j < |\mathcal{T}_u|} (t_{j+1} - t_j) \leq H.$*

**Remarks on the assumptions.** Assumption 1, 2 and 3 are standard in the domain of non-convex optimization. Comparing with the 1-bit Adam paper (Tang et al., 2021), we do not explicitly assume the uniform lower bound on the variance coordinate, i.e., $\boldsymbol{e}_j^\top \boldsymbol{v} > v_{\min} > 0, \forall j$ for some constant $v_{\min}$. Instead we assume an infinity-norm bound on the gradient as in Assumption 3 which is more realistic. Assumption 4 is also assumed in (Tang et al., 2021), in the appendix we discuss the variant of **0/1 Adam** that converges with weaker condition on the $\mathcal{C}$.

The convergence for Algorithm 1 is then given in the follow theorem.

**Theorem 1.** *Under Assumption 1 to 5, let $m = |\mathcal{T}_v|$, select $\beta_1, \beta_2 \in [0, 1)$ that fulfills $m \leq \log(1 - \beta_1)/\log(\beta_2)$, if we run Algorithm 1 with a constant learning rate: for all $t \geq 0$*

$$\gamma_t = \min\left\{\sqrt{\frac{n}{\sigma^2 T}}, \frac{1}{4L\sqrt{G_\infty^2 + \epsilon}}, \frac{2\sqrt{G_\infty^2 + \epsilon}}{L}, \frac{1}{6}\right\},$$

*then it holds that*

$$\frac{1}{T}\sum_{t=0}^{T-1}\mathbb{E}\|\nabla f(\tilde{\boldsymbol{x}}_t)\|^2 \leq O\left(\frac{\sigma}{\sqrt{nT}} + \frac{H^2\Delta^2(m+n)}{T} + \frac{1}{T}\right),$$

*where $\tilde{\boldsymbol{x}}_t = 1/n\sum_{i=1}^n \boldsymbol{x}_t^{(i)}$ and we omit $f(\boldsymbol{0}) - \inf_{\boldsymbol{x} \in \mathbb{R}^d} f(\boldsymbol{x})$, $G_\infty$, $d$, $\epsilon$, $\beta_1$, $\beta_2$ and $L$ as constants.*

Theorem 1 shows that **0/1 Adam** Algorithm 1 essentially admits the same convergence rate as distributed SGD in the sense that it achieves linear speed up, at rate $1/O(\sqrt{nT})$. The effect of compression ($\Delta$) and local steps ($H$) only appears on a non-dominating term.

# 6 EXPERIMENTS

In this section we evaluate the performance of **0/1 Adam** over several large-scale model training tasks comparing with baselines (1-bit Adam (Tang et al., 2021) and original Adam (Kingma and Ba, 2014)). Since Tang et al. (2021) already demonstrated that 1-bit Adam has similar statistical results to Adam, we omit the comparison of end-to-end model accuracy to Adam for brevity. Throughout the experiments, we enable FP16 training for all the tasks following (Tang et al., 2021). That makes the full-precision communication (including Adam, full-precision stage in 1-bit Adam and full-precision AllReduce in **0/1 Adam**) use 16-bit per number. We use the 1-bit compressor (Equation (4)) in **0/1 Adam**.

**Experimental details.** We adopt the following tasks for the evaluation: BERT-Base ($L = 12$, $H = 768$, $A = 12$, $110M$ params) and BERT-Large ($L = 24$, $H = 1024$, $A = 16$, $340M$ params) pre-training, training Resnet18 ($12M$ params) on ImageNet (He et al., 2016) and GPT-2 pre-training. For BERT model, we use the same dataset as (Devlin et al., 2018), which is a concatenation of Wikipedia and BooksCorpus with 2.5B and 800M words respectively. We use the GLUE fine-tuning benchmark (Wang et al., 2018b) to evaluate the convergence of the BERT models trained by different algorithms. For ImageNet, we adopt ImageNet-1k dataset, which contains 1.28M images for training and 50K images

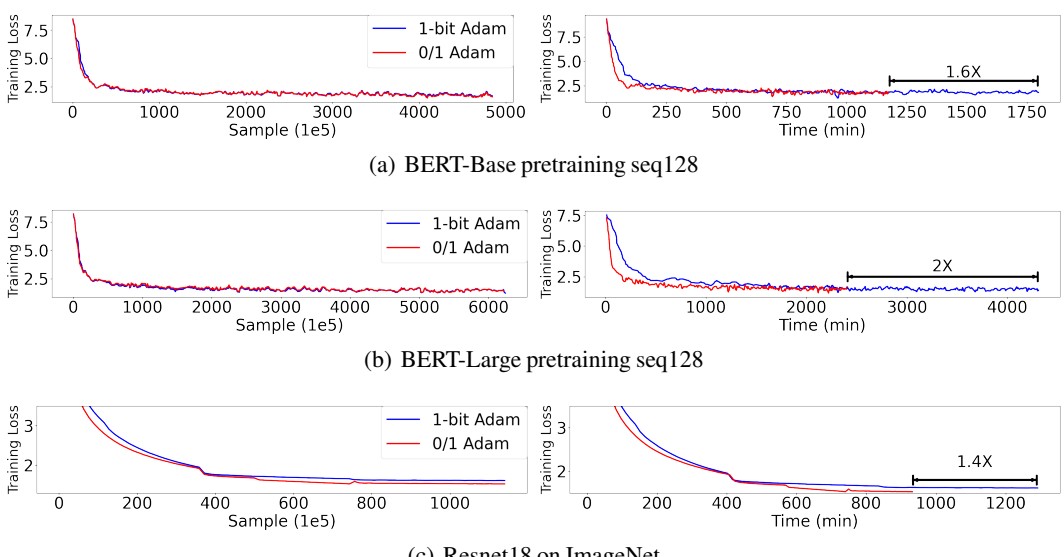

Figure 2: Sample-wise and time-wise convergence for BERT-Base/Large pre-training sequence length 128 and Resnet18 pretraining on ImageNet using 128 GPUs on the Ethernet cluster. Note that the time on the right side is measured on both algorithms processing the same number of samples.

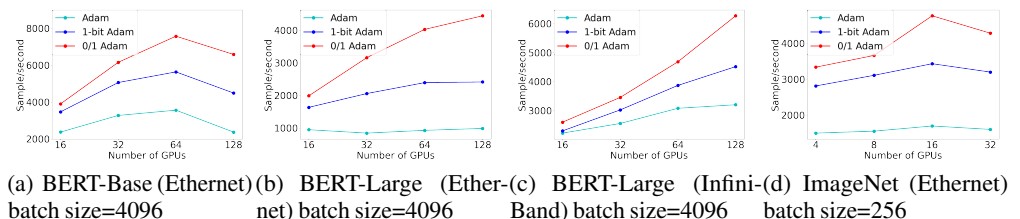

(a) BERT-Base (Ethernet) batch size=4096 (b) BERT-Large (Ethernet) batch size=4096 (c) BERT-Large (Infini-Band) batch size=4096 (d) ImageNet (Ethernet) batch size=256

Figure 3: End-to-end average throughput for BERT-Base/Large pre-training sequence length 128 and Resnet18 pretraining on ImageNet using 128 V100 GPUs on the Ethernet/InfiniBand cluster. Note that since for ImageNet, both batch size (256) and model (Resnet18) are small compared to BERT, and its parallelism speed up will be limited if applied to the same large system on BERT (128 GPUs). And so we test it for 4 to 32 GPUs in Figure (d).

for validation (Deng et al., 2009). For GPT-2 we adopt the model from its original paper (Radford et al., 2019b), which contains 117M parameters (48 layers, 1600 hidden size, 25 attention heads). For training data, we adopt the same dataset blend as in (Shoeybi et al., 2019): Wikipedia (Devlin et al., 2018), CC-Stories (Trinh and Le, 2018), RealNews (Zellers et al., 2019), and OpenWebtext (Radford et al., 2019b). Other details including learning rate schedules, hyperparameters can be found in Appendix C.

**Hardware.** We evaluate two clusters: one with 4 NVIDIA V100 GPUs per node and 40 Gigabit Ethernet inter-node network (2.7 Gbps effective bandwidth); the other one with 8 V100 GPUs per node and 100 Gigabit InfiniBand EDR inter-node network (close to theoretical peak effective bandwidth). We use 4 to 128 GPUs for BERT and ImageNet pretraining tasks to measure **0/1 Adam**'s performance gain. We use 64 GPUs for GPT-2 pre-training. Additionally, for ImageNet training we apply the accelerated data loading technique from lmdb[6].

**Policy for $\mathcal{T}_v$ and $\mathcal{T}_u$ in 0/1 Adam.** We first illustrate our policy on $\mathcal{T}_v$. Observing from our motivation study (Figure 1) that the variance difference in adjacent steps decreases roughly exponentially. Denote $k_j$ as the step where $j$-th variance update takes place, we select $\mathcal{T}_v$ such that, $k_{j+1} - k_j = 2^{\lfloor j/\kappa \rfloor}, \forall \kappa > 0$. We adopt $\kappa = 16$ for all the three tasks.

---

[6] https://github.com/xunge/pytorch_lmdb_imagenet

Table 1: GLUE development set results. BERT-Base/Large(original) results are from (Devlin et al., 2018). BERT-Base/Large(Adam and 1-bit Adam) results are from (Tang et al., 2021). The scores are the median scores over 10 runs with different seeds, and are obtained on the checkpoints trained by both sequence 128 and sequence 512 datasets.

|  | RTE | MRPC | STS-B | CoLA | SST-2 | QNLI | QQP | MNLI-(m/mm) | Avg |
|---|---|---|---|---|---|---|---|---|---|
| BERT-Base(Original) | 66.4 | 84.8 | 85.8 | 52.1 | 93.5 | 90.5 | 89.2 | 84.6/83.4 | 81.1 |
| BERT-Base(Adam) | 68.2 | 84.8 | 85.1 | 56.8 | 91.8 | 90.9 | 90.9 | 83.6/83.5 | 81.8 |
| BERT-Base(1-bit Adam) | 69.0 | 84.8 | 83.6 | 55.6 | 91.6 | 90.8 | 90.9 | 83.6/83.9 | 81.5 |
| BERT-Base(**0/1 Adam**) | 69.7 | 85.1 | 84.9 | 54.4 | 91.9 | 90.3 | 90.7 | 83.7/83.7 | 81.6 |
| BERT-Large(Original) | 70.1 | 85.4 | 86.5 | 60.5 | 94.9 | 92.7 | 89.3 | 86.7/85.9 | 83.6 |
| BERT-Large(Adam) | 70.3 | 86.0 | 86.9 | 60.3 | 93.1 | 92.2 | 91.4 | 86.1/86.2 | 83.6 |
| BERT-Large(1-bit Adam) | 70.4 | 86.1 | 86.1 | 62.0 | 93.8 | 91.9 | 91.5 | 85.7/85.4 | 83.7 |
| BERT-Large(**0/1 Adam**) | 71.7 | 86.2 | 86.9 | 59.9 | 93.2 | 91.6 | 91.4 | 85.6/85.6 | 83.6 |

Table 2: The first column shows Top1 accuracy on ImageNet of Resnet at the end of epoch 90 from different algorithms. The original accuracy is provided by Pytorch pretrained model library (Pytorch, 2014). For the other two algorithms, the accuracy is the highest score over 3 runs. The other two columns shows zero-shot evaluation of the trained GPT-2 on WikiText-103 and LAMBADA datasets, the evaluation methodology follows (Shoeybi et al., 2019). The number for Adam is obtained from (Li et al., 2021b).

|  | ImageNet Top1 Acc. $\uparrow$ | WikiText Perplexity $\downarrow$ | LAMBADA Acc. $\uparrow$ |
|---|---|---|---|
| Original Adam | 69.76 | 27.78 | 33.19 |
| 1-bit Adam | 69.93 | 28.37 | 33.21 |
| **0/1 Adam** | 69.88 | 28.07 | 33.51 |

Then we move on to discuss the policy for $\mathcal{T}_{\boldsymbol{u}}$. Based on the derivation in Section 4, the approximation noise from local step is proportional to the learning rate. And so if we denote $t_j$ as the step where $j$-th synchronization takes place, then our intuition is to increase $t_{j+1} - t_j$ roughly inversely proportional to the learning rate at $t_j$ so as to make the approximation noise bounded. For BERT-Base/Large pretraining, as illustrated before, the learning rate exponentially decreases by 0.99 every 520 steps after 12.5K linear increase warmup steps. So that we set $t_{j+1} - t_j = 1$ for the first 12.5K steps and after that let it multiply by 2 every 32678 steps based on the calculation that the learning rate will decrease by half. Similarly, for ImageNet we set $t_{j+1} - t_j = 1$ for the first 50050 steps (10 epochs) and after that let it multiply by 2 every 50050 steps (10 epochs). We clip the interval at 16 in all the tasks. This corresponds to $H = 16$ in Assumption 5. Finally, since our theory in Section 5 indicates that approximation will be more accurate when the variance is frozen. So that we additionally stop updating variance when $t_{j+1} - t_j > 1$.

**Remarks on the selected policy.** As described, the policies for $\mathcal{T}_{\boldsymbol{v}}$ and $\mathcal{T}_{\boldsymbol{u}}$ generally follow the learning rate schedule adopted. This is favored in practice for three reasons: (1) It does not require too much hyperparameter tuning. Consider adopting a constant or decaying policy based on some utility function, capturing the dynamics of the training to reach a useful interval would require tedious searching on the hyperparameters and retraining; (2) Learning rate policy is well-motivated. As naturally, communication is a way of eliminating the difference among workers. If workers adopt a larger learning rate, more frequent communication will be needed since the workers are taking large steps in the weight space and so we require higher-frequent communication for them to reach consensus; (3) Learning rate schedule is universally used in all the ML/DL applications, and thus the method can be easily adapted to other applications. In fact, if we consider the scope of large model training, a typical learning rate schedule is a linear warm-up followed by a decaying phase, which is very similar to our test cases here.

## 6.1 CONVERGENCE SPEED AND QUALITY ANALYSIS

Figure 2 presents the sample-wise and time-wise convergence results for different algorithms with 128 GPUs on the Ethernet cluster. We find that **0/1 Adam** provides the same sample-wise convergence speed compared to the baseline, with up to $2\times$ time-wise speed up. Table 1 summarizes the GLUE results using the checkpoints from our BERT pretraining experiments. **0/1 Adam** achieves similar end task accuracy compared to the numbers reported in previous work. **0/1 Adam** achieves faster training time than prior work because it reduces the communication overhead in distributed training

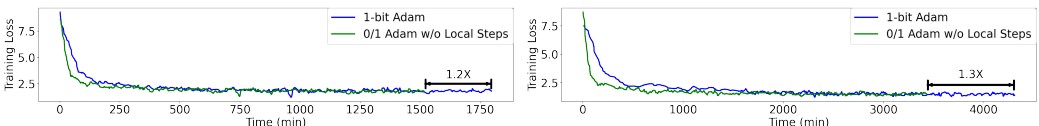

Figure 4: Reduction on number of bits per parameter used and number of communication rounds in different tasks. Note that the communication round numbers are normalized due to scale difference in different tasks. Note that original Adam uses 16bits per parameter and communicate at evert step.

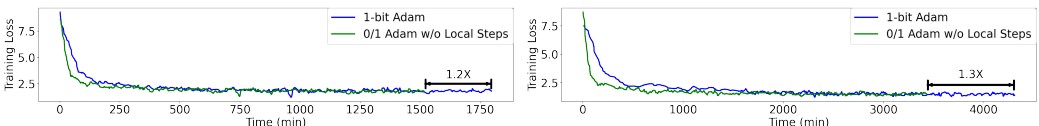

Figure 5: Evaluation BERT-Base/Large pretraining throughput using **0/1 Adam** without communication rounds skipping. Comparing with Figure 4 and 2, local steps breaks the barrier on the performance gain.

by using both 1-bit quantizer to compress the communication volume (up to $32\times$ reduction) and 1-bit AllReduce to reduce the expensive synchronization overhead for local steps in both warmup and non-warmup phases. Table 2 provides the ImageNet validation accuracy of trained models from different algorithms, and we find the final accuracy can achieve the reported accuracy from Pytorch library (Pytorch, 2014). For brevity, convergence comparison on GPT-2 is given in the Appendix C.

## 6.2 TRAINING THROUGHPUT ANALYSIS

Figure 3 summarizes the throughput results on different tasks and different clusters. We observe that **0/1 Adam** can consistently outperform baselines in all settings. It is also worth mentioning that **0/1 Adam** on Ethernet (2.7 Gbps effective bandwidth, 4 GPUs per node) is able to achieve comparable throughput as 1-bit Adam on InfiniBand (near 100 Gbps effective bandwidth, 8 GPUs per node), as shown in the red line in Figure 3(b) and the blue line in Figure 3(c), which demonstrates **0/1 Adam** further removes the redundancy in communication effectively that exceeds the hardware barrier.

**Communication reduction and the role of local steps.** To better understand the importance and effect of local steps, we additionally run a special case of **0/1 Adam** where we keep the same policy of $\mathcal{T}_{\boldsymbol{v}}$ but use $\mathcal{T}_{\boldsymbol{u}} = \{0, \cdots, T-1\}$. This special version of **0/1 Adam** does not skip rounds but use the same variance freezing policy. We plot the data volume usage and throughput results in Figure 4 and 5, respectively. We see that although no local steps suffice to reduce the data volume overhead from 1-bit Adam towards 1-bit-per-parameter in general, the throughput improvement is limited compared to Figure 2.

## 7 CONCLUSION

In this paper, we study the challenges of using 1-bit communication on Adam, and limitations of the state-of-the-art 1-bit Adam algorithm. We propose an algorithm named **0/1 Adam** that adopts two novel design: adaptive variance state freezing and 1-bit sync. We provide convergence proof for **0/1 Adam** and measure its effectiveness over baseline Adam and 1-bit Adam on various benchmarks, including BERT-Base/Large, GPT-2 pretraining and ImageNet.

### ACKNOWLEDGMENTS

Yucheng Lu is supported by Meta PhD Fellowship. The authors would like to thank anonymous reviewers from ICLR2023 for providing valuable feedback.

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

## A FULL DESCRIPTION TO ALLREDUCE

As introduced in Section 2, the error feedback based **1bit-AllReduce** works best both in theory and in practice. In fact, the original 1-bit Adam also adopts the error-feedback design (Tang et al., 2021). We give the full description of this **1bit-AllReduce** in Algorithm 2, to replace the **1bit-AllReduce** in Algorithm 1. In the theoretical analysis, our proofs will also rely on this algorithm. Note that this algorithm does not require any additional assumptions for our theory to hold, since this fits the black-box procedure in Algorithm 4 and Algorithm 1.

---

**Algorithm 2** The full description of Error Feedback 1 bit Communication (**1bit-AllReduce**)

---

**Require:** communication buffer $z_t^{(i)}$, worker error $\delta_t^{(i)}$, server error $\overline{\delta}_t$, 1-bit compressor $\mathcal{C}[\cdot]$. Both worker and server errors will be initialized at $\mathbf{0}$ at $t = 0$.

1: **(On $i$-th node)**
2: Compress $z_t^{(i)}$ into $\hat{z}_t^{(i)} = \mathcal{C}[z_t^{(i)} + \delta_t^{(i)}]$, and update the compression error by $\delta_{t+1}^{(i)} = z_t^{(i)} + \delta_t^{(i)} - \hat{z}_t^{(i)}$.
3: Send $\hat{z}_t^{(i)}$ to the server.
4: **(On server)**
5: Take the average over all the $\hat{z}_t^{(i)}$ and compress it into $\overline{z}_t = \mathcal{C}[\frac{1}{n}\sum_{i=1}^{n}\hat{z}_{t+1}^{(i)} + \overline{\delta}_t]$, and update the compression error by $\overline{\delta}_{t+1} = \frac{1}{n}\sum_{i=1}^{n}\hat{z}_t^{(i)} + \overline{\delta}_t - \overline{z}_t$.
6: Send $\overline{z}_t$ to all the workers.
7: **(On $i$-th node)**
8: **return** $\overline{z}_t, \delta_{t+1}^{(i)}, \overline{\delta}_{t+1}$.

---

---

**Algorithm 3** The full description of **AllReduce**

---

**Require:** communication buffer $z_t^{(i)}$.
 1: **(On $i$-th node)**
 2: Send $z_t^{(i)}$ to the server.
 3: **(On server)**
 4: Take the average over all the $z_t^{(i)}$ into $\overline{z}_t = \frac{1}{n}\sum_{i=1}^n z_t^{(i)}$.
 5: Send $\overline{z}_t$ to all the workers.
 6: **(On $i$-th node)**
 7: **return** $\overline{z}_t$.

---

## B   PROFILING RESULTS FOR FIXED COST OF COMMUNICATION

We profile the time taken in computation and others (including initialization of a communication round and compression) during one 1-bit AllReduce round at different scales on Ethernet cluster in Table 3.

Table 3: Profiling on Ethernet cluster the time taken in computation and others (including initialization of a communication round and compression) during one 1-bit AllReduce round at different scales.

| **ImageNet** | 4 node (16 GPUs) | 8 node (32 GPUs) | 16 node (64 GPUs) | 32 node (128 GPUs) |
|---|---|---|---|---|
| Computation | 73ms | 68ms | 44ms | 51ms |
| Others | 8ms | 6ms | 21ms | 19ms |
| **BERT-Base** | 4 node (16 GPUs) | 8 node (32 GPUs) | 16 node (64 GPUs) | 32 node (128 GPUs) |
| Computation | 941ms | 490ms | 263ms | 162ms |
| Others | 153ms | 250ms | 397ms | 658ms |
| **BERT-Large** | 4 node (16 GPUs) | 8 node (32 GPUs) | 16 node (64 GPUs) | 32 node (128 GPUs) |
| Computation | 1840ms | 970ms | 640ms | 332ms |
| Others | 340ms | 510ms | 590ms | 931ms |

## C   ADDITIONAL EXPERIMENTAL DETAILS

**Training Parameters.** For BERT pretraining, we follow the settings from (Devlin et al., 2018) and let the learning rate linearly increases to $4 \times 10^{-4}$ as a warmup in the first 12.5K steps, then decays into 0.99 of the original after every 520 steps. We set $\beta_1 = 0.9$ and $\beta_2 = 0.999$ for all the algorithms. We adopt the batch size of 4096. For 1-bit Adam, we follow the hyperparameters given in (Tang et al., 2021) and set the full-precision stage for 1-bit Adam as 16K and 23K on BERT-Base and BERT-Large, respectively. All the hyperparameters used here (e.g. learning rate) strictly follow (Tang et al., 2021) for fair comparison. For ImageNet, we follow the example script from Pytorch[7] and use batch size of 256 and a milestone decay learning rate schedule: starting at 1e-4 and decay by a factor of 10 at epoch 30 and 60, with 90 epochs in total. We set 10 epochs (50050 steps) as the full-precision stage for 1-bit Adam. For GPT-2 we set batch size to be 512, and use 300K training steps (158B tokens). The learning rate schedule follows a linear warmup of 3K steps and a single cycle consine decay over the remaining 297K steps ($1 \times 10^{-5}$min). For 1-bit Adam, we set its full-precision stage length to be 80K steps, and for the **0/1 Adam**, we follow the same learning rate based policy from BERT on $\mathcal{T}_v$ and $\mathcal{T}_u$.

For GLUE benchmarks we use Adam optimizer and perform single-task training on the dev set. Following the setup in the BERT paper (Devlin et al., 2018) and 1-bit Adam paper (Tang et al., 2021), we search over the hyperparameter space with batch sizes $\in \{8, 16, 32\}$ and learning rates $\{1 \times 10^{-5}, 3 \times 10^{-5}, 5 \times 10^{-5}, 8 \times 10^{-5}\}$.

The convergence plots for GPT-2 pre-training are given in Figure 6.

---

[7]`https://github.com/pytorch/examples/blob/master/imagenet/main.py`

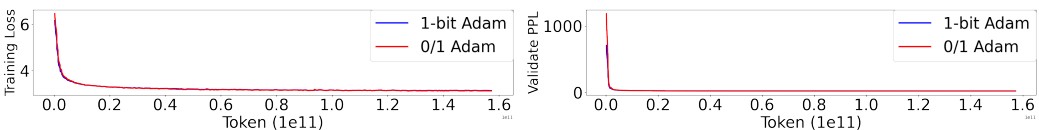

Figure 6: Training loss (left) and validation perplexity (right) with respect to Tokens for 1-bit Adam and **0/1 Adam**.

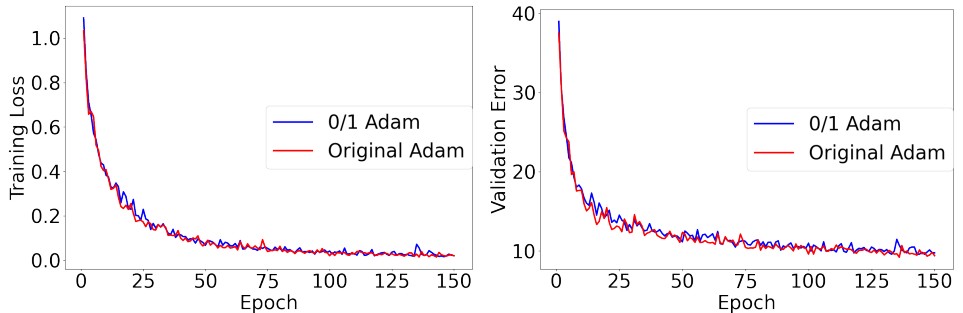

Figure 7: Training loss and Validation error of ResNet18-CIFAR10 task. Hyperparameters that are not related to **0/1 Adam** is set to be {learning rate: $1e-4$, weight decay: $5e-4$}. For hyperparameters associated with **0/1 Adam**, we use the same ones as used in ImageNet with no additional tuning.

## D   THEORETICAL ANALYSIS

### D.1   ANALYSIS TO AN INTERMEDIATE VERSION OF **0/1 ADAM**

---

**Algorithm 4** Generic framework of applying 1-bit communication to Adam with frozen variance state. 1-bit Adam can be viewed as a special case of setting $\mathcal{T}_v = \{0, \cdots, T_0 - 1\}$ where $T_0$ denotes its total number of steps in the full-precision stage.

---

**Require:** initialized model on worker $i$: $\boldsymbol{x}_0^{(i)}$, learning rate $\{\gamma_t\}_{t=1}^T$, $\boldsymbol{m}_0 = \boldsymbol{0}$, $\boldsymbol{v}_0 = \boldsymbol{0}$, total number of iterations $T$, decaying factor $\beta_1, \beta_2$ from Adam, numerical constant $\epsilon$, variance update step index set $\mathcal{T}_v$.

1: **for** $t = 0, \cdots, T-1$ **do**
2:       Locally compute stochastic gradient $\boldsymbol{g}_t^{(i)}$ over $\boldsymbol{x}_t^{(i)}$.
3:       **if** $t \in \mathcal{T}_v$ **then**
4:             $\overline{\boldsymbol{g}}_t = \textbf{AllReduce}\left(\boldsymbol{g}_t^{(i)}\right)$.
5:             Set $\boldsymbol{v}_{t+1} = \beta_2 \boldsymbol{v}_t + (1 - \beta_2)(\overline{\boldsymbol{g}}_t)^2$.
6:       **else**
7:             $\overline{\boldsymbol{g}}_t = \textbf{Compressed-AllReduce}\left(\boldsymbol{g}_t^{(i)}\right)$.
8:             Set $\boldsymbol{v}_{t+1} = \boldsymbol{v}_t$.
9:       **end if**
10:      Update momentum: $\boldsymbol{m}_{t+1} = \beta_1 \boldsymbol{m}_t + (1 - \beta_1)\overline{\boldsymbol{g}}_t$.
11:      Update model: $\boldsymbol{x}_{t+1}^{(i)} = \boldsymbol{x}_t^{(i)} - \gamma_t \boldsymbol{m}_t / \sqrt{\boldsymbol{v}_t + \epsilon}$.
12: **end for**
13: **return** $\boldsymbol{x}_T^{(i)}, \forall i$.

---

We start from a special case of **0/1 Adam** that compresses gradients without local steps. This is given in Algorithm 4. Note that the following proof will use Algorithm 2 to replace **Compressed-AllReduce** in Algorithm 4, as introduced in Section A.

Algorithm 4 allows us to work with weaker assumption as given in the following

**Assumption 6.** *Compression error in Algorithm 4: For arbitrary $\boldsymbol{x} \in \mathbb{R}^d$, there exists a constant $0 \leq \omega < 1$, such that the output of compressor $\mathcal{C}[\cdot]$ has the following error bound:*

$$\mathbb{E}\|\mathcal{C}[\boldsymbol{x}] - \boldsymbol{x}\|^2 \leq \omega \|\boldsymbol{x}\|^2.$$

**Theorem 2.** *Under Assumption 1, 2, 3, and 6, let $m = |\mathcal{T}_{\boldsymbol{v}}|$, and select $\beta_1, \beta_2 \in [0, 1)$ such that $m \leq \log(1 - \beta_1)/\log(\beta_2)$. If we run Algorithm 4 with a constant learning rate: for all $t \geq 0$*

$$\gamma_t = \min\left\{ \sqrt{\frac{n}{\sigma^2 T}}, \frac{1}{2L\sqrt{G_\infty^2 + \epsilon}}, \frac{1}{125} \right\},$$

*then it holds that*

$$\frac{1}{T}\sum_{t=0}^{T-1} \mathbb{E}\|\nabla f(\boldsymbol{x}_t)\|^2 \leq O\left( \frac{\sigma}{\sqrt{nT}} + \frac{m+n}{(1-\omega)^4 T} + \frac{1}{T} \right),$$

*where we omit $f(\mathbf{0}) - \inf_{\boldsymbol{x} \in \mathbb{R}^d} f(\boldsymbol{x})$, $G_\infty$, $d$, $\epsilon$, $\beta_1$, $\beta_2$ and $L$ as constants.*

*Proof.* The main update of Algorithm 4 (with constant learning rate) can be summarized as: for every $t = 0, \cdots, T-1$,

$$\boldsymbol{m}_{t+1} = \beta_1 \boldsymbol{m}_t + (1-\beta_1)\overline{\boldsymbol{g}}_t$$

$$\boldsymbol{v}_{t+1} = \begin{cases} \beta_2 \boldsymbol{v}_t + (1-\beta_2)\left(\frac{1}{n}\sum_{i=1}^n \boldsymbol{g}_t^{(i)}\right)^2 & t \in \mathcal{T}_{\boldsymbol{v}}, \\ \\ \boldsymbol{v}_t & t \notin \mathcal{T}_{\boldsymbol{v}}. \end{cases}$$

$$\boldsymbol{x}_{t+1} = \boldsymbol{x}_t - \gamma \frac{\boldsymbol{m}_t}{\sqrt{\boldsymbol{v}_t + \epsilon}},$$

where the $\overline{\boldsymbol{g}}_t$ is the output of the **1-bit AllReduce** algorithm[8]. Note that based on Algorithm 2, the gradient approximation term follows:

$$\overline{\boldsymbol{g}}_t = \frac{1}{n}\sum_{i=1}^n \hat{\boldsymbol{g}}_t^{(i)} + \overline{\boldsymbol{\delta}}_t - \overline{\boldsymbol{\delta}}_{t+1}$$

$$= \frac{1}{n}\sum_{i=1}^n \left( \boldsymbol{g}_t^{(i)} + \boldsymbol{\delta}_t^{(i)} - \boldsymbol{\delta}_{t+1}^{(i)} \right) + \overline{\boldsymbol{\delta}}_t - \overline{\boldsymbol{\delta}}_{t+1}$$

$$= \frac{1}{n}\sum_{i=1}^n \boldsymbol{g}_t^{(i)} + \left( \frac{1}{n}\sum_{i=1}^n \boldsymbol{\delta}_t^{(i)} - \overline{\boldsymbol{\delta}}_t \right) - \left( \frac{1}{n}\sum_{i=1}^n \boldsymbol{\delta}_{t+1}^{(i)} - \overline{\boldsymbol{\delta}}_{t+1} \right)$$

$$= \boldsymbol{g}_t + \boldsymbol{\delta}_t - \boldsymbol{\delta}_{t+1},$$

where we denote

$$\boldsymbol{g}_t = \frac{1}{n}\sum_{i=1}^n \boldsymbol{g}_t^{(i)}$$

$$\boldsymbol{\delta}_t = \frac{1}{n}\sum_{i=1}^n \boldsymbol{\delta}_t^{(i)} - \overline{\boldsymbol{\delta}}_t.$$

To prove the convergence, we now define the following auxiliary sequence: for any $t \geq 0$,

$$\boldsymbol{y}_t = \boldsymbol{x}_t - \frac{\gamma \boldsymbol{m}_t}{(1-\beta_1)\sqrt{\boldsymbol{v}_t + \epsilon}} - \frac{\gamma \boldsymbol{\delta}_t}{\sqrt{\boldsymbol{v}_t + \epsilon}}.$$

The rest of the proof is to use this auxiliary sequence to bound two types of steps separately. We call a step $t$ as *reuse step* if $t \notin \mathcal{T}_{\boldsymbol{v}}$ and *update step* otherwise. We see for all the update steps, $\boldsymbol{v}_t \neq \boldsymbol{v}_{t+1}$ while for all the reuse steps $\boldsymbol{v}_t = \boldsymbol{v}_{t+1}$. The bounds on two different types of steps are provides by Lemma 5 and Lemma 6. Specifically, denoting $V_1 = \left\| \frac{1}{\sqrt{\boldsymbol{v}_1 + \epsilon}} \right\|_1$, from Lemma 5 we obtain for all the reuse steps,

$$\sum_{t \notin \mathcal{T}_{\boldsymbol{v}}} \frac{\gamma}{4\sqrt{G_\infty^2 + \epsilon}} \mathbb{E}\|\nabla f(\boldsymbol{x}_t)\|^2$$

$$\leq \sum_{t \notin \mathcal{T}_{\boldsymbol{v}}} \mathbb{E}[f(\boldsymbol{y}_t) - f(\boldsymbol{y}_{t+1})] + \frac{227\gamma^3 L^2 V_1^2 (1+\omega)^3 G_\infty^2 d\sqrt{G_\infty^2 + \epsilon}(T-m)}{\beta_2^{2m}(1-\beta_1)^2(1-\omega)^4} + \frac{L\gamma^2\sigma^2 V_1(T-m)}{2n\beta_2^m}.$$

---

[8]In the original Algorithm 4, the $\overline{\boldsymbol{g}}_t$ is the output of the **AllReduce** when $t \in \mathcal{T}_{\boldsymbol{v}}$. This, however, does not affect our analysis, since our proof holds for a noisier case. The original Algorithm 4 is mainly for practical concern – we avoid redundant **AllReduce** rounds when **1-bit AllReduce** is performed.

while from Lemma 6 we obtain for all the update steps,

$$\sum_{t\in\mathcal{T}_v}\frac{\gamma}{4\sqrt{G_\infty^2+\epsilon}}\mathbb{E}\|\nabla f(\boldsymbol{x}_t)\|^2\leq\sum_{t\in\mathcal{T}_v}\mathbb{E}[f(\boldsymbol{y}_t)-f(\boldsymbol{y}_{t+1})]+\left(\frac{34\gamma}{L}+\frac{\gamma}{4\sqrt{G_\infty^2+\epsilon}}\right)\cdot\left(\frac{\sigma^2}{n}+G_\infty^2 d\right)m$$
$$+\frac{32\gamma(1+\beta_1)^2(1+\omega)^3 V_1 G_\infty^2 dmL}{\beta_2^m(1-\beta_1)^2(1-\omega)^4}.$$

Note that the two inequalities above hold when the learning rate fulfills

$$\gamma\leq\min\left\{\frac{\beta_2^m}{2V_1 L\sqrt{G_\infty^2+\epsilon}},\frac{1}{125}\right\}.$$

Combine them together,

$$\frac{1}{T}\sum_{t=0}^{T-1}\frac{\mathbb{E}\|\nabla f(\boldsymbol{x}_t)\|^2}{4\sqrt{G_\infty^2+\epsilon}}\leq\frac{f(\boldsymbol{0})-f^*}{\gamma T}+\frac{227\gamma^2 L^2 V_1^2(1+\omega)^3 G_\infty^2 d\sqrt{G_\infty^2+\epsilon}(T-m)}{\beta_2^{2m}(1-\beta_1)^2(1-\omega)^4 T}+\frac{L\gamma\sigma^2 V_1(T-m)}{2n\beta_2^m T}$$
$$+\left(\frac{34}{L}+\frac{1}{4\sqrt{G_\infty^2+\epsilon}}\right)\cdot\left(\frac{\sigma^2}{n}+G_\infty^2 d\right)\frac{m}{T}+\frac{32(1+\beta_1)^2(1+\omega)^3 V_1 G_\infty^2 dmL}{\beta_2^m(1-\beta_1)^2(1-\omega)^4 T}$$

Dropping the constants, we finally obtain

$$\frac{1}{T}\sum_{t=0}^{T-1}\mathbb{E}\|\nabla f(\boldsymbol{x}_t)\|^2\leq O\left(\frac{f(\boldsymbol{0})-f^*}{\gamma T}+\frac{\gamma^2}{\beta_2^{2m}(1-\beta_1)^2(1-\omega)^4}+\frac{\gamma\sigma^2}{n\beta_2^m}+\frac{\omega m}{\beta_2^m(1-\beta_1)^2(1-\omega)^4 T}+\frac{\sigma^2 m}{nT}\right)$$
$$\leq O\left(\frac{f(\boldsymbol{0})-f^*}{\gamma T}+\frac{\gamma^2}{(1-\beta_1)^4(1-\omega)^4}+\frac{\gamma\sigma^2}{n(1-\beta_1)}+\frac{\omega m}{(1-\beta_1)^3(1-\omega)^4 T}+\frac{\sigma^2 m}{nT}\right),$$

where in the last step we use the condition in the theorem that $\beta_2^m\geq 1-\beta_1$. To meet the requirement of learning rate we set

$$\gamma_t=\min\left\{\sqrt{\frac{n}{\sigma^2 T}},\frac{1}{2L\sqrt{G_\infty^2+\epsilon}},\frac{1}{125}\right\},$$

then it holds that

$$\frac{1}{T}\sum_{t=0}^{T-1}\mathbb{E}\|\nabla f(\boldsymbol{x}_t)\|^2\leq O\left(\frac{\sigma}{\sqrt{nT}}+\frac{m+n}{(1-\omega)^4 T}+\frac{1}{T}\right).$$

That completes the proof. $\qquad\square$

### D.2 PROOF TO THEOREM 1

Note that the following proof will use Algorithm 2 to replace **1bit-AllReduce** in Algorithm 1, as introduced in Section A.

**Theorem 1.** *Under Assumption 1 to 5, let* $m=|\mathcal{T}_v|$, *select* $\beta_1$, $\beta_2\in[0,1)$ *that fulfills* $m\leq\log(1-\beta_1)/\log(\beta_2)$, *if we run Algorithm 1 with a constant learning rate: for all* $t\geq 0$

$$\gamma_t=\min\left\{\sqrt{\frac{n}{\sigma^2 T}},\frac{1}{4L\sqrt{G_\infty^2+\epsilon}},\frac{2\sqrt{G_\infty^2+\epsilon}}{L},\frac{1}{6}\right\},$$

*then it holds that*

$$\frac{1}{T}\sum_{t=0}^{T-1}\mathbb{E}\|\nabla f(\tilde{\boldsymbol{x}}_t)\|^2\leq O\left(\frac{\sigma}{\sqrt{nT}}+\frac{H^2\Delta^2(m+n)}{T}+\frac{1}{T}\right),$$

*where* $\tilde{\boldsymbol{x}}_t=1/n\sum_{i=1}^n \boldsymbol{x}_t^{(i)}$ *and we omit* $f(\boldsymbol{0})-\inf_{\boldsymbol{x}\in\mathbb{R}^d}f(\boldsymbol{x})$, $G_\infty$, $d$, $\epsilon$, $\beta_1$, $\beta_2$ *and* $L$ *as constants.*

*Proof.* We now prove Theorem 1. Similar to the proof to Theorem 2, in this proof we discuss the case of $t\in\mathcal{T}_v$ and $t\notin\mathcal{T}_v$ separately. Following the proof of Theorem 2, we define the following auxiliary sequence

$$\tilde{\boldsymbol{y}}_t=\tilde{\boldsymbol{x}}_t-\frac{\gamma\tilde{\boldsymbol{m}}_t}{(1-\beta_1)\sqrt{\boldsymbol{v}_t+\epsilon}}-\frac{\gamma\boldsymbol{\delta}_t}{\sqrt{\boldsymbol{v}_t+\epsilon}},$$

where

$$\tilde{\boldsymbol{x}}_t=\frac{1}{n}\sum_{i=1}^n \boldsymbol{x}_t^{(i)}$$

$$\tilde{m}_t = \frac{1}{n}\sum_{i=1}^{n} m_t^{(i)}.$$

And we additionally define that

$$\tilde{u}_t = \frac{1}{n}\sum_{i=1}^{n} u_t^{(i)}$$

$$\tilde{g}_t = \frac{1}{n}\sum_{i=1}^{n} g_t^{(i)}.$$

Note that the definition of $\tilde{g}_t$ is different from the $g_t$ in Theorem 2 since the former is computed on local models which potentially can be different before the sync step.

To expect a compression error bound to scale in the order of $O(\gamma^2)$, we slightly modify the update of line 5, 8, 9 of Algorithm 1 into

$$u_{t+\frac{1}{2}}^{(i)} = u_t^{(i)} + m_t^{(i)}$$

$$m_{t+1}^{(i)} = \overline{u}_{t+\frac{1}{2}} / \sum_{k=t'}^{t}$$

$$x_{t+1}^{(i)} = x_{t'}^{(i)} - \gamma \overline{u}_{t+\frac{1}{2}} / \sqrt{v_t + \epsilon}.$$

Note that since Theorem 1 states the convergence results for constant learning rate, such modification does not change the semantics of the original Algorithm 1. Based on Algorithm 2, we know that

$$\overline{u}_{t+\frac{1}{2}} = \frac{1}{n}\sum_{i=1}^{n} \hat{u}_{t+\frac{1}{2}}^{(i)} + \overline{\delta}_t - \overline{\delta}_{t+1}$$

$$= \frac{1}{n}\sum_{i=1}^{n} \left( u_{t+\frac{1}{2}}^{(i)} + \delta_t^{(i)} - \delta_{t+1}^{(i)} \right) + \overline{\delta}_t - \overline{\delta}_{t+1}$$

$$= \frac{1}{n}\sum_{i=1}^{n} u_{t+\frac{1}{2}}^{(i)} + \left( \frac{1}{n}\sum_{i=1}^{n} \delta_t^{(i)} - \overline{\delta}_t \right) - \left( \frac{1}{n}\sum_{i=1}^{n} \delta_{t+1}^{(i)} - \overline{\delta}_{t+1} \right)$$

$$= \tilde{u}_{t+\frac{1}{2}} + \delta_t - \delta_{t+1}.$$

Based on Lemma 10, we know that for all the $t \in \mathcal{T}_v$, we have the following bound,

$$\sum_{t\in\mathcal{T}_v} \frac{\gamma \mathbb{E}\|\nabla f(\tilde{x}_t)\|^2}{4\sqrt{G_\infty^2 + \epsilon}} \leq \sum_{t\in\mathcal{T}_v} \mathbb{E}f(\tilde{y}_t) - \mathbb{E}f(\tilde{y}_{t+1}) + \frac{2\gamma\sigma^2 m}{nL} + \frac{106\gamma H^2 V_1(M+\Delta^2)mL}{\beta_2^m(1-\beta_1)^2}$$

$$+ \frac{\gamma\sigma^2 m}{4n\sqrt{G_\infty^2+\epsilon}} + \frac{\gamma G_\infty^2 dm}{4\sqrt{G_\infty^2+\epsilon}}.$$

On the other hand, for all the $t \notin \mathcal{T}_v$, we have the following bound,

$$\sum_{t\notin\mathcal{T}_v} \frac{\gamma \mathbb{E}\|\nabla f(\tilde{x}_t)\|^2}{4\sqrt{G_\infty^2 + \epsilon}}$$

$$\leq \sum_{t\notin\mathcal{T}_v} \mathbb{E}f(\tilde{y}_t) - \mathbb{E}f(\tilde{y}_{t+1}) + \frac{36\gamma^3 H^2 V_1(3G_\infty^2 d + 25\Delta^2)L^2(1+L)(G_\infty^2+\epsilon+1)(T-m)}{\beta_2^m(1-\beta_1)^4\sqrt{G_\infty^2+\epsilon}}$$

$$+ \frac{L\gamma^2 V_1 \sigma^2(T-m)}{n\beta_2^m} + \frac{48\gamma^3 V_1(H+1)^2(3G_\infty^2 d + 24\Delta^2)\sqrt{G_\infty^2+\epsilon}(T-m)}{\beta_2^m(1-\beta_1)^4}.$$

Note that they hold if learning rate is set to be

$$\gamma \leq \min\left\{ \frac{\beta_2^m}{4V_1 L\sqrt{G_\infty^2+\epsilon}}, \frac{2\sqrt{G_\infty^2+\epsilon}}{L}, \frac{1}{6} \right\}.$$

Combine them together, we obtain

$$\frac{1}{T}\sum_{t=0}^{T-1} \frac{\mathbb{E}\|\nabla f(x_t)\|^2}{4\sqrt{G_\infty^2+\epsilon}}$$

$$\leq \frac{f(\mathbf{0})-f^*}{\gamma T} + \frac{2\sigma^2 m}{nLT} + \frac{106 H^2 V_1(M+\Delta^2)mL}{\beta_2^m(1-\beta_1)^2 T} + \frac{\sigma^2 m}{4n\sqrt{G_\infty^2+\epsilon}T} + \frac{G_\infty^2 dm}{4\sqrt{G_\infty^2+\epsilon}T} + \frac{L\gamma V_1\sigma^2}{n\beta_2^m}$$

$$+\frac{36\gamma^2 H^2 V_1(3G_\infty^2 d+25\Delta^2)L^2(1+L)(G_\infty^2+\epsilon+1)}{\beta_2^m(1-\beta_1)^4\sqrt{G_\infty^2+\epsilon}}$$

$$+\frac{48\gamma^2 V_1(H+1)^2(3G_\infty^2 d+24\Delta^2)\sqrt{G_\infty^2+\epsilon}}{\beta_2^m(1-\beta_1)^4}.$$

Omitting constants:

$$\frac{1}{T}\sum_{t=0}^{T-1}\mathbb{E}\|\nabla f(\tilde{\boldsymbol{x}}_t)\|^2\leq O\left(\frac{f(\mathbf{0})-f^*}{\gamma T}+\frac{\gamma^2 H^2\Delta^2}{\beta_2^m}+\frac{\gamma\sigma^2}{n\beta_2^m}+\frac{\sigma^2 m}{nT}+\frac{H^2\Delta^2 m}{\beta_2^m T}+\frac{m}{T}\right)$$

$$\leq O\left(\frac{f(\mathbf{0})-f^*}{\gamma T}+\frac{\gamma^2 H^2\Delta^2}{1-\beta_1}+\frac{\gamma\sigma^2}{n(1-\beta_1)}+\frac{\sigma^2 m}{nT}+\frac{H^2\Delta^2 m}{(1-\beta_1)T}+\frac{m}{T}\right),$$

where in the last step we use the condition in the theorem that $\beta_2^m\geq 1-\beta_1$. To meet the requirement of learning rate we set

$$\gamma_t=\min\left\{\sqrt{\frac{n}{\sigma^2 T}},\frac{1}{4L\sqrt{G_\infty^2+\epsilon}},\frac{2\sqrt{G_\infty^2+\epsilon}}{L},\frac{1}{6}\right\},$$

then it holds that

$$\frac{1}{T}\sum_{t=0}^{T-1}\mathbb{E}\|\nabla f(\tilde{\boldsymbol{x}}_t)\|^2\leq O\left(\frac{\sigma}{\sqrt{nT}}+\frac{H^2\Delta^2(m+n)}{T}+\frac{1}{T}\right).$$

And that completes the proof. $\qquad\square$

### D.3 TECHNICAL LEMMA

**Lemma 1.** *Consider running Algorithm 2 over a communication buffer $\boldsymbol{z}$ (same notation in Algorithm 2) under Assumption 6, let $\boldsymbol{\delta}_t$ denote:*

$$\boldsymbol{\delta}_t=\frac{1}{n}\sum_{i=1}^n\boldsymbol{\delta}_t^{(i)}-\overline{\boldsymbol{\delta}}_t$$

*then based on Assumption 6 and 3, it holds that $t\geq 0$, if $\mathbb{E}\|\boldsymbol{z}_t^{(i)}\|^2\leq C$ for some constant $C>0$,*

$$\mathbb{E}\|\boldsymbol{\delta}_t\|^2\leq\frac{32\omega(1+\omega)^3 C}{(1-\omega)^4}.$$

*Proof.* Note that the error is initialized by $\mathbf{0}$, so that when $t=0$ the bound trivially holds. We next prove the case for $t\geq 1$.

For any $i\in\{1,\cdots,n\}$ and $t\geq 1$, by the definition of the sequence $\boldsymbol{\delta}_t^{(i)}$,

$$\mathbb{E}\left\|\boldsymbol{\delta}_t^{(i)}\right\|^2=\mathbb{E}\left\|\boldsymbol{z}_{t-1}^{(i)}+\boldsymbol{\delta}_{t-1}^{(i)}-\hat{\boldsymbol{z}}_{t-1}^{(i)}\right\|^2$$

$$=\mathbb{E}\left\|\boldsymbol{z}_{t-1}^{(i)}+\boldsymbol{\delta}_{t-1}^{(i)}-\mathcal{C}\left[\boldsymbol{z}_{t-1}^{(i)}+\boldsymbol{\delta}_{t-1}^{(i)}\right]\right\|^2$$

$$\overset{Assumption\ 6}{\leq}\omega\mathbb{E}\left\|\boldsymbol{z}_{t-1}^{(i)}+\boldsymbol{\delta}_{t-1}^{(i)}\right\|^2$$

$$\overset{\forall\eta>0}{=}\omega(1+\eta)\mathbb{E}\left\|\boldsymbol{\delta}_{t-1}^{(i)}\right\|^2+\omega(1+1/\eta)\mathbb{E}\left\|\boldsymbol{z}_{t-1}^{(i)}\right\|^2$$

$$\overset{Assumption\ 3}{\leq}\sum_{j=0}^\infty[\omega(1+\eta)]^j\omega(1+1/\eta)C$$

$$\leq\frac{\omega(1+1/\eta)}{1-\omega(1+\eta)}C.$$

Selecting $\eta=\frac{1-\omega}{2\omega}$, we obtain

$$\mathbb{E}\left\|\boldsymbol{\delta}_t^{(i)}\right\|^2\leq\frac{2\omega(1+\omega)}{(1-\omega)^2}C.$$

Similarly, we can show that for any $t\geq 1$,

$$\mathbb{E}\left\|\overline{\boldsymbol{\delta}}_t\right\|^2=\mathbb{E}\left\|\frac{1}{n}\sum_{i=1}^n\hat{\boldsymbol{z}}_{t-1}^{(i)}+\overline{\boldsymbol{\delta}}_{t-1}-\overline{\boldsymbol{z}}_{t-1}\right\|^2$$

$$=\mathbb{E}\left\|\frac{1}{n}\sum_{i=1}^{n}\hat{\boldsymbol{z}}_{t-1}^{(i)}+\overline{\boldsymbol{\delta}}_{t-1}-\mathcal{C}\left[\frac{1}{n}\sum_{i=1}^{n}\hat{\boldsymbol{z}}_{t-1}^{(i)}+\overline{\boldsymbol{\delta}}_{t-1}\right]\right\|^{2}$$

$$\leq\omega\mathbb{E}\left\|\frac{1}{n}\sum_{i=1}^{n}\hat{\boldsymbol{z}}_{t-1}^{(i)}+\overline{\boldsymbol{\delta}}_{t-1}\right\|^{2}$$

$$\leq\omega(1+\eta)\mathbb{E}\left\|\overline{\boldsymbol{\delta}}_{t-1}\right\|^{2}+\omega(1+1/\eta)\mathbb{E}\left\|\frac{1}{n}\sum_{i=1}^{n}\hat{\boldsymbol{z}}_{t-1}^{(i)}\right\|^{2}$$

$$\leq\omega(1+\eta)\mathbb{E}\left\|\overline{\boldsymbol{\delta}}_{t-1}\right\|^{2}+\omega(1+1/\eta)\cdot\frac{1}{n}\sum_{i=1}^{n}\mathbb{E}\left\|\hat{\boldsymbol{z}}_{t-1}^{(i)}\right\|^{2},$$

where in the last step we apply the Jensen Inequality. Since we do not assume a bound on the $\left\|\hat{\boldsymbol{z}}_{t-1}^{(i)}\right\|^{2}$, we need to bound it in terms of

$$\mathbb{E}\left\|\hat{\boldsymbol{z}}_{t-1}^{(i)}\right\|^{2}=\mathbb{E}\left\|\boldsymbol{z}_{t-1}^{(i)}+\boldsymbol{\delta}_{t-1}^{(i)}-\boldsymbol{\delta}_{t}^{(i)}\right\|^{2}$$

$$\leq2\mathbb{E}\left\|\boldsymbol{z}_{t-1}^{(i)}+\boldsymbol{\delta}_{t-1}^{(i)}\right\|^{2}+2\mathbb{E}\left\|\boldsymbol{\delta}_{t}^{(i)}\right\|^{2}$$

$$\leq\frac{4(1+\omega)}{(1-\omega)^{2}}C+\frac{4\omega(1+\omega)}{(1-\omega)^{2}}C$$

$$\leq\frac{4(1+\omega)^{2}}{(1-\omega)^{2}}C,$$

where we apply the results from the bound on $\mathbb{E}\left\|\boldsymbol{\delta}_{t}^{(i)}\right\|^{2}$. Given this bound, and following the analysis for $\mathbb{E}\left\|\boldsymbol{\delta}_{t}^{(i)}\right\|^{2}$, we can now bound the $\mathbb{E}\left\|\overline{\boldsymbol{\delta}}_{t}\right\|^{2}$ as follows

$$\mathbb{E}\left\|\overline{\boldsymbol{\delta}}_{t}\right\|^{2}\leq\frac{2\omega(1+\omega)}{(1-\omega)^{2}}\cdot\frac{4(1+\omega)^{2}}{(1-\omega)^{2}}C$$

$$=\frac{8\omega(1+\omega)^{3}}{(1-\omega)^{4}}C.$$

Finally, we obtain $t\geq1$,

$$\mathbb{E}\left\|\boldsymbol{\delta}_{t}\right\|^{2}=\mathbb{E}\left\|\frac{1}{n}\sum_{i=1}^{n}\boldsymbol{\delta}_{t}^{(i)}-\overline{\boldsymbol{\delta}}_{t}\right\|^{2}$$

$$\leq2\mathbb{E}\left\|\overline{\boldsymbol{\delta}}_{t}\right\|^{2}+2\mathbb{E}\left\|\frac{1}{n}\sum_{i=1}^{n}\boldsymbol{\delta}_{t}^{(i)}\right\|^{2}$$

$$\leq2\mathbb{E}\left\|\overline{\boldsymbol{\delta}}_{t}\right\|^{2}+2\frac{1}{n}\sum_{i=1}^{n}\mathbb{E}\left\|\boldsymbol{\delta}_{t}^{(i)}\right\|^{2}$$

$$\leq\frac{32\omega(1+\omega)^{3}C}{(1-\omega)^{4}}.$$

That completes the proof. $\square$

**Lemma 2.** *For the variance term, we have the following upper and lower bound: for any $t\geq1$,*

$$\beta_{2}^{m/2}\sqrt{\boldsymbol{v}_{1}+\epsilon}\leq\sqrt{\boldsymbol{v}_{t}+\epsilon}\leq\sqrt{G_{\infty}^{2}+\epsilon},$$

*where the inequality holds element-wise.*

*Proof.* On one hand, for any $t_{j}\leq t<t_{j+1}$, where $t_{j}$ denotes an update step, we obtain element-wise:

$$\boldsymbol{v}_{t}\geq\beta_{2}\boldsymbol{v}_{t_{j}}\geq\cdots\geq\beta_{2}^{j}\boldsymbol{v}_{1}\geq\beta_{2}^{m}\boldsymbol{v}_{1},$$

so that

$$\sqrt{\boldsymbol{v}_{t}+\epsilon}\geq\sqrt{\beta_{2}^{m}\boldsymbol{v}_{1}+\epsilon}\geq\sqrt{\beta_{2}^{m}(\boldsymbol{v}_{1}+\epsilon)}=\beta_{2}^{m/2}\sqrt{\boldsymbol{v}_{1}+\epsilon}.$$

On the other hand, for any $t \geq 1$ and $j \in \{1, \cdots, d\}$,

$$[\boldsymbol{v}_t]_j = \sum_{k=1}^{t}(1-\beta_2)\beta_2^{t-k}\left(\frac{1}{n}\sum_{i=1}^{n}[\boldsymbol{g}_k^{(i)}]_j\right)^2 \leq G_\infty^2(1-\beta_2)\sum_{k=1}^{\infty}\beta_2^k \leq G_\infty^2,$$

so that

$$\sqrt{\boldsymbol{v}_t + \epsilon} \leq \sqrt{G_\infty^2 + \epsilon}.$$

That completes the proof. $\qquad\square$

**Lemma 3.** *In Algorithm 4, for any $t \geq 0$,*

$$\mathbb{E}\|\boldsymbol{m}_t\|^2 \leq \frac{195(1+\omega)^3 G_\infty^2 d}{(1-\omega)^4}.$$

*Proof.* For any $t \geq 0$,

$$\mathbb{E}\|\boldsymbol{m}_t\|^2 = \mathbb{E}\left\|(1-\beta_1)\sum_{k=0}^{t}\beta_1^{t-k}\overline{\boldsymbol{g}}_k\right\|^2$$

$$\leq (1-\beta_1)\sum_{k=0}^{t}\beta_1^{t-k}\mathbb{E}\|\overline{\boldsymbol{g}}_k\|^2$$

$$\leq (1-\beta_1)\sum_{k=0}^{t}\beta_1^{t-k}\mathbb{E}\|\boldsymbol{g}_k+\boldsymbol{\delta}_k-\boldsymbol{\delta}_{k+1}\|^2$$

$$\leq (1-\beta_1)\sum_{k=0}^{t}\beta_1^{t-k}\left(3\mathbb{E}\|\boldsymbol{g}_k\|^2+3\mathbb{E}\|\boldsymbol{\delta}_k\|^2+3\mathbb{E}\|\boldsymbol{\delta}_{k+1}\|^2\right)$$

$$\leq (1-\beta_1)\sum_{k=0}^{t}\beta_1^{t-k}\left(3\mathbb{E}\left\|\frac{1}{n}\sum_{i=1}^{n}\boldsymbol{g}_k^{(i)}\right\|^2+3\mathbb{E}\|\boldsymbol{\delta}_k\|^2+3\mathbb{E}\|\boldsymbol{\delta}_{k+1}\|^2\right)$$

$$\leq (1-\beta_1)\sum_{k=0}^{t}\beta_1^{t-k}\left(\frac{3}{n}\sum_{i=1}^{n}\mathbb{E}\left\|\boldsymbol{g}_k^{(i)}\right\|^2+3\mathbb{E}\|\boldsymbol{\delta}_k\|^2+3\mathbb{E}\|\boldsymbol{\delta}_{k+1}\|^2\right)$$

$$\overset{(i)}{\leq} (1-\beta_1)\sum_{k=0}^{t}\beta_1^{t-k}\left(3G_\infty^2 d+\frac{192\omega(1+\omega)^3 G_\infty^2 d}{(1-\omega)^4}\right)$$

$$\leq \left(\frac{3(1+\omega)^3 G_\infty^2 d}{(1-\omega)^4}+\frac{192(1+\omega)^3 G_\infty^2 d}{(1-\omega)^4}\right)\cdot(1-\beta_1)\sum_{k=0}^{\infty}\beta_1^k$$

$$\leq \frac{195(1+\omega)^3 G_\infty^2 d}{(1-\omega)^4},$$

where in the step (i) we use Lemma 1. That completes the proof. $\qquad\square$

**Lemma 4.** *For any $\boldsymbol{a}, \boldsymbol{b} \in \mathbb{R}^d$, the following bound holds:*

$$\left\|\frac{\boldsymbol{a}}{\sqrt{\boldsymbol{b}}}\right\|^2 \leq \|\boldsymbol{a}\|^2\left\|\frac{1}{\boldsymbol{b}}\right\|_1.$$

*Proof.* Denote the subscript $j$ as the index of the coordinate.

$$\left\|\frac{\boldsymbol{a}}{\sqrt{\boldsymbol{b}}}\right\|^2 = \sum_{j=1}^{d}\left(\frac{\boldsymbol{a}_j}{[\sqrt{\boldsymbol{b}}]_j}\right)^2 \leq \left(\sum_{j=1}^{d}\boldsymbol{a}_j^2\right)\left(\sum_{j=1}^{d}\frac{1}{\boldsymbol{b}_j}\right) = \left(\sum_{j=1}^{d}\boldsymbol{a}_j^2\right)\left(\sum_{j=1}^{d}\left|\frac{1}{\boldsymbol{b}_j}\right|\right) = \|\boldsymbol{a}\|^2\left\|\frac{1}{\boldsymbol{b}}\right\|_1.$$

Note that the second step holds not because Cauchy-Schwarz Inequality but due to the fact that $\boldsymbol{a}_j^2$, $\boldsymbol{b}_j > 0$ (since $\sqrt{\boldsymbol{b}}$ would implicitly assume so). $\qquad\square$

**Lemma 5.** *In Algorithm 4, for all the $t \geq 1$ that fulfills $\boldsymbol{v}_t = \boldsymbol{v}_{t+1}$, i.e., $\forall t$ such that $t \notin \mathcal{T}_{\boldsymbol{v}}$, if we let*

$$\gamma \leq \frac{\beta_2^m}{2V_1 L\sqrt{G_\infty^2+\epsilon}},$$

*the following bound holds,*

$$\sum_{t\notin\mathcal{T}_{\boldsymbol{v}}}\frac{\gamma}{4\sqrt{G_\infty^2+\epsilon}}\mathbb{E}\|\nabla f(\boldsymbol{x}_t)\|^2$$

$$\leq\sum_{t\notin\mathcal{T}_{\boldsymbol{v}}}\mathbb{E}[f(\boldsymbol{y}_t)-f(\boldsymbol{y}_{t+1})]+\frac{227\gamma^3L^2V_1^2(1+\omega)^3G_\infty^2d\sqrt{G_\infty^2+\epsilon}(T-m)}{\beta_2^{2m}(1-\beta_1)^2(1-\omega)^4}+\frac{L\gamma^2\sigma^2V_1(T-m)}{2n\beta_2^m}.$$

*Proof.* Recall the auxiliary sequence

$$\boldsymbol{y}_t=\boldsymbol{x}_t-\frac{\gamma\boldsymbol{m}_t}{(1-\beta_1)\sqrt{\boldsymbol{v}_t+\epsilon}}-\frac{\gamma\boldsymbol{\delta}_t}{\sqrt{\boldsymbol{v}_t+\epsilon}},$$

For all the steps $t\geq0$ that fulfills $\boldsymbol{v}_{t+1}=\boldsymbol{v}_t$, we obtain

$$\boldsymbol{y}_{t+1}-\boldsymbol{y}_t=\boldsymbol{x}_{t+1}-\boldsymbol{x}_t-\frac{\gamma}{1-\beta_1}\left(\frac{\boldsymbol{m}_{t+1}}{\sqrt{\boldsymbol{v}_{t+1}+\epsilon}}-\frac{\boldsymbol{m}_t}{\sqrt{\boldsymbol{v}_t+\epsilon}}\right)-\gamma\left(\frac{\boldsymbol{\delta}_{t+1}}{\sqrt{\boldsymbol{v}_{t+1}+\epsilon}}-\frac{\boldsymbol{\delta}_t}{\sqrt{\boldsymbol{v}_t+\epsilon}}\right)$$

$$=-\gamma\frac{\boldsymbol{m}_t}{\sqrt{\boldsymbol{v}_t+\epsilon}}-\frac{\gamma}{(1-\beta_1)\sqrt{\boldsymbol{v}_t+\epsilon}}(\beta_1\boldsymbol{m}_t+(1-\beta_1)\overline{\boldsymbol{g}}_t-\boldsymbol{m}_t-(1-\beta_1)(\boldsymbol{\delta}_t-\boldsymbol{\delta}_{t+1}))$$

$$=-\frac{\gamma\boldsymbol{g}_t}{\sqrt{\boldsymbol{v}_t+\epsilon}}.$$

From Assumption 1, we have

$$\mathbb{E}f(\boldsymbol{y}_{t+1})-\mathbb{E}f(\boldsymbol{y}_t)$$

$$\leq\mathbb{E}\langle\nabla f(\boldsymbol{y}_t),\boldsymbol{y}_{t+1}-\boldsymbol{y}_t\rangle+\frac{L}{2}\mathbb{E}\|\boldsymbol{y}_{t+1}-\boldsymbol{y}_t\|^2$$

$$=-\gamma\mathbb{E}\left\langle\nabla f(\boldsymbol{y}_t),\frac{\boldsymbol{g}_t}{\sqrt{\boldsymbol{v}_t+\epsilon}}\right\rangle+\frac{L\gamma^2}{2}\mathbb{E}\left\|\frac{\boldsymbol{g}_t}{\sqrt{\boldsymbol{v}_t+\epsilon}}\right\|^2$$

$$=-\gamma\mathbb{E}\left\langle\nabla f(\boldsymbol{y}_t),\frac{\nabla f(\boldsymbol{x}_t)}{\sqrt{\boldsymbol{v}_t+\epsilon}}\right\rangle+\frac{L\gamma^2}{2}\mathbb{E}\left\|\frac{\boldsymbol{g}_t}{\sqrt{\boldsymbol{v}_t+\epsilon}}\right\|^2$$

$$=-\gamma\mathbb{E}\left\langle\nabla f(\boldsymbol{x}_t),\frac{\nabla f(\boldsymbol{x}_t)}{\sqrt{\boldsymbol{v}_t+\epsilon}}\right\rangle+\gamma\mathbb{E}\left\langle\nabla f(\boldsymbol{x}_t)-\nabla f(\boldsymbol{y}_t),\frac{\nabla f(\boldsymbol{x}_t)}{\sqrt{\boldsymbol{v}_t+\epsilon}}\right\rangle+\frac{L\gamma^2}{2}\mathbb{E}\left\|\frac{\boldsymbol{g}_t}{\sqrt{\boldsymbol{v}_t+\epsilon}}\right\|^2$$

$$=-\gamma\mathbb{E}\left\langle\nabla f(\boldsymbol{x}_t),\frac{\nabla f(\boldsymbol{x}_t)}{\sqrt{\boldsymbol{v}_t+\epsilon}}\right\rangle+\gamma\mathbb{E}\left\langle\frac{\nabla f(\boldsymbol{x}_t)-\nabla f(\boldsymbol{y}_t)}{\sqrt{\boldsymbol{v}_t+\epsilon}},\nabla f(\boldsymbol{x}_t)\right\rangle+\frac{L\gamma^2}{2}\mathbb{E}\left\|\frac{\boldsymbol{g}_t}{\sqrt{\boldsymbol{v}_t+\epsilon}}\right\|^2$$

$$\leq-\frac{\gamma\mathbb{E}\|\nabla f(\boldsymbol{x}_t)\|^2}{\sqrt{G_\infty^2+\epsilon}}+\frac{\gamma}{2\eta}\mathbb{E}\left\|\frac{\nabla f(\boldsymbol{x}_t)-\nabla f(\boldsymbol{y}_t)}{\sqrt{\boldsymbol{v}_t+\epsilon}}\right\|^2+\frac{\gamma\eta}{2}\mathbb{E}\|\nabla f(\boldsymbol{x}_t)\|^2+\frac{L\gamma^2}{2}\mathbb{E}\left\|\frac{\boldsymbol{g}_t}{\sqrt{\boldsymbol{v}_t+\epsilon}}\right\|^2,$$

where in the last step we use Lemma 2 and the fact that for any $\boldsymbol{a},\boldsymbol{b}$ and constant $\eta>0$,

$$\langle\boldsymbol{a},\boldsymbol{b}\rangle\leq\frac{\eta}{2}\|\boldsymbol{a}\|^2+\frac{1}{2\eta}\|\boldsymbol{b}\|^2.$$

Set $\eta=(\sqrt{G_\infty^2+\epsilon})^{-1}$, with Assumption 1 and Lemma 4,

$$\mathbb{E}f(\boldsymbol{y}_{t+1})-\mathbb{E}f(\boldsymbol{y}_t)$$

$$\leq-\frac{\gamma\mathbb{E}\|\nabla f(\boldsymbol{x}_t)\|^2}{2\sqrt{G_\infty^2+\epsilon}}+\frac{\gamma L^2V_1\sqrt{G_\infty^2+\epsilon}}{2\beta_2^m}\mathbb{E}\|\boldsymbol{x}_t-\boldsymbol{y}_t\|^2+\frac{L\gamma^2}{2}\mathbb{E}\left\|\frac{\boldsymbol{g}_t}{\sqrt{\boldsymbol{v}_t+\epsilon}}\right\|^2$$

$$=-\frac{\gamma\mathbb{E}\|\nabla f(\boldsymbol{x}_t)\|^2}{2\sqrt{G_\infty^2+\epsilon}}+\frac{\gamma L^2V_1\sqrt{G_\infty^2+\epsilon}}{2\beta_2^m}\mathbb{E}\left\|\frac{\gamma\boldsymbol{m}_t}{(1-\beta_1)\sqrt{\boldsymbol{v}_t+\epsilon}}+\frac{\gamma\boldsymbol{\delta}_t}{\sqrt{\boldsymbol{v}_t+\epsilon}}\right\|^2+\frac{L\gamma^2}{2}\mathbb{E}\left\|\frac{\boldsymbol{g}_t}{\sqrt{\boldsymbol{v}_t+\epsilon}}\right\|^2$$

$$\leq-\frac{\gamma\mathbb{E}\|\nabla f(\boldsymbol{x}_t)\|^2}{2\sqrt{G_\infty^2+\epsilon}}+\frac{\gamma^3L^2V_1\sqrt{G_\infty^2+\epsilon}}{\beta_2^m(1-\beta_1)^2}\mathbb{E}\left\|\frac{\boldsymbol{m}_t}{\sqrt{\boldsymbol{v}_t+\epsilon}}\right\|^2+\frac{\gamma^3L^2V_1\sqrt{G_\infty^2+\epsilon}}{\beta_2^m}\mathbb{E}\left\|\frac{\boldsymbol{\delta}_t}{\sqrt{\boldsymbol{v}_t+\epsilon}}\right\|^2$$

$$+\frac{L\gamma^2}{2}\mathbb{E}\left\|\frac{\boldsymbol{g}_t}{\sqrt{\boldsymbol{v}_t+\epsilon}}\right\|^2$$

$$\leq-\frac{\gamma\mathbb{E}\|\nabla f(\boldsymbol{x}_t)\|^2}{2\sqrt{G_\infty^2+\epsilon}}+\frac{\gamma^3L^2V_1^2\sqrt{G_\infty^2+\epsilon}}{\beta_2^{2m}(1-\beta_1)^2}\mathbb{E}\|\boldsymbol{m}_t\|^2+\frac{\gamma^3L^2V_1^2\sqrt{G_\infty^2+\epsilon}}{\beta_2^{2m}}\mathbb{E}\|\boldsymbol{\delta}_t\|^2+\frac{L\gamma^2V_1}{2\beta_2^m}\mathbb{E}\|\boldsymbol{g}_t\|^2,$$

where in the last step we apply Lemma 2 and 4. Using the bound on the error from Lemma 1, Lemma 3 and the assumption on the stochastic gradient, we obtain

$$\left(\frac{\gamma}{2\sqrt{G_\infty^2+\epsilon}}-\frac{L\gamma^2 V_1}{2\beta_2^m}\right)\mathbb{E}\|\nabla f(\boldsymbol{x}_t)\|^2$$

$$\leq\mathbb{E}[f(\boldsymbol{y}_t)-f(\boldsymbol{y}_{t+1})]+\frac{\gamma^3 L^2 V_1^2\sqrt{G_\infty^2+\epsilon}}{\beta_2^{2m}(1-\beta_1)^2}\mathbb{E}\|\boldsymbol{m}_t\|^2+\frac{\gamma^3 L^2 V_1^2\sqrt{G_\infty^2+\epsilon}}{\beta_2^{2m}}\mathbb{E}\|\boldsymbol{\delta}_t\|^2+\frac{L\gamma^2\sigma^2 V_1}{2n\beta_2^m}$$

$$\leq\mathbb{E}[f(\boldsymbol{y}_t)-f(\boldsymbol{y}_{t+1})]+\frac{195\gamma^3 L^2 V_1^2(1+\omega)^3 G_\infty^2 d\sqrt{G_\infty^2+\epsilon}}{\beta_2^{2m}(1-\beta_1)^2(1-\omega)^4}+\frac{32\gamma^3 L^2 V_1^2\omega(1+\omega)^3 G_\infty^2 d\sqrt{G_\infty^2+\epsilon}}{\beta_2^{2m}(1-\omega)^4}$$

$$+\frac{L\gamma^2\sigma^2 V_1}{2n\beta_2^m}$$

$$\leq\mathbb{E}[f(\boldsymbol{y}_t)-f(\boldsymbol{y}_{t+1})]+\frac{227\gamma^3 L^2 V_1^2(1+\omega)^3 G_\infty^2 d\sqrt{G_\infty^2+\epsilon}}{\beta_2^{2m}(1-\beta_1)^2(1-\omega)^4}+\frac{L\gamma^2\sigma^2 V_1}{2n\beta_2^m}.$$

Based on the learning rate bound

$$\gamma\leq\frac{\beta_2^m}{2V_1 L\sqrt{G_\infty^2+\epsilon}},$$

and summing over all the reuse steps, we obtain

$$\sum_{t\notin\mathcal{T}_{\boldsymbol{v}}}\frac{\gamma}{4\sqrt{G_\infty^2+\epsilon}}\mathbb{E}\|\nabla f(\boldsymbol{x}_t)\|^2$$

$$\leq\sum_{t\notin\mathcal{T}_{\boldsymbol{v}}}\mathbb{E}[f(\boldsymbol{y}_t)-f(\boldsymbol{y}_{t+1})]+\frac{227\gamma^3 L^2 V_1^2(1+\omega)^3 G_\infty^2 d\sqrt{G_\infty^2+\epsilon}(T-m)}{\beta_2^{2m}(1-\beta_1)^2(1-\omega)^4}+\frac{L\gamma^2\sigma^2 V_1(T-m)}{2n\beta_2^m}.$$

That completes the proof. □

**Lemma 6.** *In Algorithm 4, for all the $t\geq 0$ that fulfills $\boldsymbol{v}_t\neq\boldsymbol{v}_{t+1}$, i.e. $t\in\mathcal{T}_{\boldsymbol{v}}$, if the learning rate fulfills*

$$\gamma<\frac{1}{125}$$

*, the following bound holds*

$$\sum_{t\in\mathcal{T}_{\boldsymbol{v}}}\frac{\gamma}{4\sqrt{G_\infty^2+\epsilon}}\mathbb{E}\|\nabla f(\boldsymbol{x}_t)\|^2\leq\sum_{t\in\mathcal{T}_{\boldsymbol{v}}}\mathbb{E}[f(\boldsymbol{y}_t)-f(\boldsymbol{y}_{t+1})]+\left(\frac{34\gamma}{L}+\frac{\gamma}{4\sqrt{G_\infty^2+\epsilon}}\right)\cdot\left(\frac{\sigma^2}{n}+G_\infty^2 d\right)m$$

$$+\frac{32\gamma(1+\beta_1)^2(1+\omega)^3 V_1 G_\infty^2 dmL}{\beta_2^m(1-\beta_1)^2(1-\omega)^4}.$$

*Proof.* For all the steps $t$ that fulfills $\boldsymbol{v}_t\neq\boldsymbol{v}_{t+1}$,

$$\boldsymbol{y}_{t+1}-\boldsymbol{y}_t=\boldsymbol{x}_{t+1}-\boldsymbol{x}_t-\frac{\gamma}{1-\beta_1}\left(\frac{\boldsymbol{m}_{t+1}}{\sqrt{\boldsymbol{v}_{t+1}+\epsilon}}-\frac{\boldsymbol{m}_t}{\sqrt{\boldsymbol{v}_t+\epsilon}}\right)+\gamma\left(\frac{\boldsymbol{\delta}_t}{\sqrt{\boldsymbol{v}_t+\epsilon}}-\frac{\boldsymbol{\delta}_{t+1}}{\sqrt{\boldsymbol{v}_{t+1}+\epsilon}}\right)$$

$$=-\gamma\frac{\boldsymbol{m}_t}{\sqrt{\boldsymbol{v}_t+\epsilon}}-\frac{\gamma}{1-\beta_1}\left(\frac{\boldsymbol{m}_{t+1}}{\sqrt{\boldsymbol{v}_{t+1}+\epsilon}}-\frac{\boldsymbol{m}_t}{\sqrt{\boldsymbol{v}_t+\epsilon}}\right)+\gamma\left(\frac{\boldsymbol{\delta}_t}{\sqrt{\boldsymbol{v}_t+\epsilon}}-\frac{\boldsymbol{\delta}_{t+1}}{\sqrt{\boldsymbol{v}_{t+1}+\epsilon}}\right)$$

$$=-\frac{\gamma\beta_1}{1-\beta_1}\frac{\boldsymbol{m}_t}{\sqrt{\boldsymbol{v}_t+\epsilon}}-\frac{\gamma}{1-\beta_1}\frac{\boldsymbol{m}_{t+1}}{\sqrt{\boldsymbol{v}_{t+1}+\epsilon}}+\gamma\left(\frac{\boldsymbol{\delta}_t}{\sqrt{\boldsymbol{v}_t+\epsilon}}-\frac{\boldsymbol{\delta}_{t+1}}{\sqrt{\boldsymbol{v}_{t+1}+\epsilon}}\right).$$

Based on the smoothness assumption, for constant $\eta>0$ that will be assigned later,

$$\mathbb{E}f(\boldsymbol{y}_{t+1})-\mathbb{E}f(\boldsymbol{y}_t)$$

$$\leq\mathbb{E}\langle\nabla f(\boldsymbol{y}_t),\boldsymbol{y}_{t+1}-\boldsymbol{y}_t\rangle+\frac{L}{2}\mathbb{E}\|\boldsymbol{y}_{t+1}-\boldsymbol{y}_t\|^2$$

$$\overset{\gamma\eta<1}{\leq}\frac{\eta\gamma}{2L}\mathbb{E}\|\nabla f(\boldsymbol{y}_t)\|^2+\frac{L}{\eta\gamma}\mathbb{E}\|\boldsymbol{y}_{t+1}-\boldsymbol{y}_t\|^2$$

$$\leq\frac{\eta\gamma}{L}\mathbb{E}\|\nabla f(\boldsymbol{x}_t)\|^2+\eta\gamma L\mathbb{E}\|\boldsymbol{y}_t-\boldsymbol{x}_t\|^2+\frac{L}{\eta\gamma}\mathbb{E}\|\boldsymbol{y}_{t+1}-\boldsymbol{y}_t\|^2$$

$$\leq\frac{\eta\gamma}{L}\mathbb{E}\|\nabla f(\boldsymbol{x}_t)-\boldsymbol{g}_t\|^2+\frac{\eta\gamma}{L}\mathbb{E}\|\boldsymbol{g}_t\|^2+\eta\gamma L\mathbb{E}\|\boldsymbol{y}_t-\boldsymbol{x}_t\|^2+\frac{L}{\eta\gamma}\mathbb{E}\|\boldsymbol{y}_{t+1}-\boldsymbol{y}_t\|^2$$

$$\leq \frac{\eta\gamma}{n^2 L}\sum_{i=1}^{n}\mathbb{E}\left\|\nabla f(\boldsymbol{x}_t)-\boldsymbol{g}_t^{(i)}\right\|^2 + \frac{\eta\gamma}{nL}\sum_{i=1}^{n}\mathbb{E}\left\|\boldsymbol{g}_t^{(i)}\right\|^2 + \eta\gamma L\mathbb{E}\|\boldsymbol{y}_t-\boldsymbol{x}_t\|^2 + \frac{L}{\eta\gamma}\mathbb{E}\|\boldsymbol{y}_{t+1}-\boldsymbol{y}_t\|^2$$

$$\leq \frac{\eta\gamma}{L}\left(\frac{\sigma^2}{n}+G_\infty^2 d\right) + \eta\gamma L\mathbb{E}\|\boldsymbol{y}_t-\boldsymbol{x}_t\|^2 + \frac{L}{\eta\gamma}\mathbb{E}\|\boldsymbol{y}_{t+1}-\boldsymbol{y}_t\|^2.$$

Now we can bound the last two terms as follows, note that

$$\mathbb{E}\|\boldsymbol{y}_t-\boldsymbol{x}_t\|^2 = \mathbb{E}\left\|\frac{\gamma\boldsymbol{m}_t}{(1-\beta_1)\sqrt{\boldsymbol{v}_t+\epsilon}} + \frac{\gamma\boldsymbol{\delta}_t}{\sqrt{\boldsymbol{v}_t+\epsilon}}\right\|^2$$

$$\leq \frac{2\gamma^2}{(1-\beta_1)^2}\mathbb{E}\left\|\frac{\boldsymbol{m}_t}{\sqrt{\boldsymbol{v}_t+\epsilon}}\right\|^2 + 2\gamma^2\mathbb{E}\left\|\frac{\boldsymbol{\delta}_t}{\sqrt{\boldsymbol{v}_t+\epsilon}}\right\|^2$$

$$\leq \frac{2\gamma^2 V_1}{(1-\beta_1)^2\beta_2^m}\mathbb{E}\|\boldsymbol{m}_t\|^2 + \frac{2\gamma^2 V_1}{\beta_2^m}\mathbb{E}\|\boldsymbol{\delta}_t\|^2$$

$$\leq \frac{390\gamma^2(1+\omega)^3 V_1 G_\infty^2 d}{\beta_2^m(1-\beta_1)^2(1-\omega)^4} + \frac{64\gamma^2\omega(1+\omega)^3 V_1 G_\infty^2 d}{\beta_2^m(1-\omega)^4}$$

$$\leq \frac{454\gamma^2(1+\omega)^3 V_1 G_\infty^2 d}{\beta_2^m(1-\beta_1)^2(1-\omega)^4},$$

where in the last step we apply Lemma 1. On the other hand,

$$\mathbb{E}\|\boldsymbol{y}_{t+1}-\boldsymbol{y}_t\|^2$$

$$=\mathbb{E}\left\|\frac{\gamma\beta_1}{1-\beta_1}\frac{\boldsymbol{m}_t}{\sqrt{\boldsymbol{v}_t+\epsilon}} + \frac{\gamma}{1-\beta_1}\frac{\boldsymbol{m}_{t+1}}{\sqrt{\boldsymbol{v}_{t+1}+\epsilon}} - \gamma\left(\frac{\boldsymbol{\delta}_t}{\sqrt{\boldsymbol{v}_t+\epsilon}} - \frac{\boldsymbol{\delta}_{t+1}}{\sqrt{\boldsymbol{v}_{t+1}+\epsilon}}\right)\right\|^2$$

$$\leq\mathbb{E}\left\|\frac{\gamma\beta_1}{1-\beta_1}\frac{\boldsymbol{m}_t}{\sqrt{\boldsymbol{v}_t+\epsilon}} + \frac{\gamma}{1-\beta_1}\frac{\boldsymbol{m}_{t+1}}{\sqrt{\boldsymbol{v}_{t+1}+\epsilon}} - \gamma\left(\frac{\boldsymbol{\delta}_t}{\sqrt{\boldsymbol{v}_t+\epsilon}} - \frac{\boldsymbol{\delta}_{t+1}}{\sqrt{\boldsymbol{v}_{t+1}+\epsilon}}\right)\right\|^2$$

$$\leq \frac{4\gamma^2\beta_1^2}{(1-\beta_1)^2}\mathbb{E}\left\|\frac{\boldsymbol{m}_t}{\sqrt{\boldsymbol{v}_t+\epsilon}}\right\|^2 + \frac{4\gamma^2}{(1-\beta_1)^2}\mathbb{E}\left\|\frac{\boldsymbol{m}_{t+1}}{\sqrt{\boldsymbol{v}_{t+1}+\epsilon}}\right\|^2 + 4\gamma^2\mathbb{E}\left\|\frac{\boldsymbol{\delta}_t}{\sqrt{\boldsymbol{v}_t+\epsilon}}\right\|^2 + 4\gamma^2\mathbb{E}\left\|\frac{\boldsymbol{\delta}_{t+1}}{\sqrt{\boldsymbol{v}_{t+1}+\epsilon}}\right\|^2$$

$$\leq \frac{4\gamma^2\beta_1^2 V_1}{(1-\beta_1)^2\beta_2^m}\mathbb{E}\|\boldsymbol{m}_t\|^2 + \frac{4\gamma^2 V_1}{(1-\beta_1)^2\beta_2^m}\mathbb{E}\|\boldsymbol{m}_{t+1}\|^2 + \frac{4\gamma^2 V_1}{\beta_2^m}\mathbb{E}\|\boldsymbol{\delta}_t\|^2 + \frac{4\gamma^2 V_1}{\beta_2^m}\mathbb{E}\|\boldsymbol{\delta}_{t+1}\|^2$$

$$\leq \frac{780\gamma^2(1+\beta_1^2)V_1(1+\omega)^3 G_\infty^2 d}{\beta_2^m(1-\beta_1)^2(1-\omega)^4} + \frac{256\gamma^2 V_1\omega(1+\omega)^3 G_\infty^2 d}{\beta_2^m(1-\omega)^4}$$

$$\leq \frac{1036\gamma^2(1+\beta_1^2)V_1(1+\omega)^3 G_\infty^2 d}{\beta_2^m(1-\beta_1)^2(1-\omega)^4},$$

where we again apply Lemma 1 and Lemma 3. Put everything together,

$$\mathbb{E}f(\boldsymbol{y}_{t+1})-\mathbb{E}f(\boldsymbol{y}_t)$$

$$\leq \frac{\eta\gamma}{L}\left(\frac{\sigma^2}{n}+G_\infty^2 d\right) + \eta\gamma L\mathbb{E}\|\boldsymbol{y}_t-\boldsymbol{x}_t\|^2 + \frac{L}{\eta\gamma}\mathbb{E}\|\boldsymbol{y}_{t+1}-\boldsymbol{y}_t\|^2$$

$$\leq \frac{\eta\gamma}{L}\left(\frac{\sigma^2}{n}+G_\infty^2 d\right) + \frac{454\eta\gamma^3(1+\omega)^3 V_1 G_\infty^2 dL}{\beta_2^m(1-\beta_1)^2(1-\omega)^4} + \frac{1036\gamma(1+\beta_1^2)V_1(1+\omega)^3 G_\infty^2 dL}{\eta\beta_2^m(1-\beta_1)^2(1-\omega)^4}$$

$$\leq \frac{\eta\gamma}{L}\left(\frac{\sigma^2}{n}+G_\infty^2 d\right) + \left(454\eta\gamma^2 + \frac{1036}{\eta}\right)\frac{\gamma(1+\beta_1)^2(1+\omega)^3 V_1 G_\infty^2 dL}{\beta_2^m(1-\beta_1)^2(1-\omega)^4}$$

Set $\eta=34$, and considering $\gamma<\frac{1}{125}$, we get

$$\mathbb{E}f(\boldsymbol{y}_{t+1})-\mathbb{E}f(\boldsymbol{y}_t) \leq \frac{34\gamma}{L}\left(\frac{\sigma^2}{n}+G_\infty^2 d\right) + \frac{32\gamma(1+\beta_1)^2(1+\omega)^3 V_1 G_\infty^2 dL}{\beta_2^m(1-\beta_1)^2(1-\omega)^4}.$$

Summing over all the update steps, we obtain

$$0 \leq \sum_{t\in\mathcal{T}_{\boldsymbol{v}}}\mathbb{E}[f(\boldsymbol{y}_t)-f(\boldsymbol{y}_{t+1})] + \frac{34\gamma}{L}\left(\frac{\sigma^2 m}{n}+G_\infty^2 dm\right) + \frac{32\gamma(1+\beta_1)^2(1+\omega)^3 V_1 G_\infty^2 dmL}{\beta_2^m(1-\beta_1)^2(1-\omega)^4}.$$

Adding $\frac{\gamma}{4\sqrt{G_\infty^2+\epsilon}}\sum_{t\in\mathcal{T}_{\boldsymbol{v}}}\mathbb{E}\|\nabla f(\boldsymbol{x}_t)\|^2$ on both sides, and note that

$$\sum_{t\in\mathcal{T}_{\boldsymbol{v}}}\mathbb{E}\|\nabla f(\boldsymbol{x}_t)\|^2 = \sum_{t\in\mathcal{T}_{\boldsymbol{v}}}\mathbb{E}\|\nabla f(\boldsymbol{x}_t)-\boldsymbol{g}_t\|^2 + \sum_{t\in\mathcal{T}_{\boldsymbol{v}}}\mathbb{E}\|\boldsymbol{g}_t\|^2$$

$$\leq \frac{\sigma^2 m}{n} + G_\infty^2 dm,$$

we finally obtain

$$\sum_{t \in \mathcal{T}_v} \frac{\gamma}{4\sqrt{G_\infty^2 + \epsilon}} \mathbb{E}\|\nabla f(\boldsymbol{x}_t)\|^2 \leq \sum_{t \in \mathcal{T}_v} \mathbb{E}[f(\boldsymbol{y}_t) - f(\boldsymbol{y}_{t+1})] + \left(\frac{34\gamma}{L} + \frac{\gamma}{4\sqrt{G_\infty^2 + \epsilon}}\right) \cdot \left(\frac{\sigma^2}{n} + G_\infty^2 d\right) m$$
$$+ \frac{32\gamma(1+\beta_1)^2(1+\omega)^3 V_1 G_\infty^2 dmL}{\beta_2^m(1-\beta_1)^2(1-\omega)^4}.$$

That completes the proof. $\qquad\square$

**Lemma 7.** *Under Assumption 4, for any $t \geq 0$, it holds that*
$$\mathbb{E}\|\boldsymbol{\delta}_t\|^2 \leq 4\Delta^2.$$

*Proof.* Based on the definition of the compression error, we obtain

$$\mathbb{E}\|\boldsymbol{\delta}_t\|^2 = \mathbb{E}\left\|\frac{1}{n}\sum_{i=1}^n \boldsymbol{\delta}_t^{(i)} - \overline{\boldsymbol{\delta}}_t\right\|^2$$
$$\leq 2\mathbb{E}\left\|\overline{\boldsymbol{\delta}}_t\right\|^2 + 2\mathbb{E}\left\|\frac{1}{n}\sum_{i=1}^n \boldsymbol{\delta}_t^{(i)}\right\|^2$$
$$\leq 2\mathbb{E}\left\|\overline{\boldsymbol{\delta}}_t\right\|^2 + 2\frac{1}{n}\sum_{i=1}^n \mathbb{E}\left\|\boldsymbol{\delta}_t^{(i)}\right\|^2$$
$$\leq 4\Delta^2.$$

That completes the proof. $\qquad\square$

**Lemma 8.** *In Algorithm 1, for any $t \geq 0$, the momentum term is uniformly bounded by the following:*
$$\mathbb{E}\left\|\boldsymbol{m}_t^{(i)}\right\|^2 \leq \frac{3G_\infty^2 d + 24\Delta^2}{(1-\beta_1)^2},$$
$$\mathbb{E}\left\|\boldsymbol{m}_{t+\frac{1}{2}}^{(i)}\right\|^2 \leq \frac{3G_\infty^2 d + 24\Delta^2}{(1-\beta_1)^2},$$
$$\mathbb{E}\|\tilde{\boldsymbol{m}}_t\|^2 \leq \frac{3G_\infty^2 d + 24\Delta^2}{(1-\beta_1)^2},$$
$$\mathbb{E}\left\|\tilde{\boldsymbol{m}}_{t+\frac{1}{2}}\right\|^2 \leq \frac{3G_\infty^2 d + 24\Delta^2}{(1-\beta_1)^2}.$$

*Proof.* We prove this lemma via induction. Note that when $t = 0$, the inequality trivially holds due to initialization at $\mathbf{0}$ and Jensen Inequality. Now suppose the inequality holds up to step $t \geq 0$, then for $t+1$, if $t \in \mathcal{T}_{\boldsymbol{u}}$, then

$$\mathbb{E}\left\|\boldsymbol{m}_{t+1}^{(i)}\right\|^2$$
$$= \mathbb{E}\left\|\frac{\overline{\boldsymbol{u}}_{t+\frac{1}{2}}}{t-k}\right\|^2$$
$$= \mathbb{E}\left\|\frac{\tilde{\boldsymbol{u}}_{t+\frac{1}{2}} + \boldsymbol{\delta}_t - \boldsymbol{\delta}_{t+1}}{t-k}\right\|^2$$
$$= \mathbb{E}\left\|\frac{\sum_{j=k+1}^t \tilde{\boldsymbol{m}}_j + \boldsymbol{\delta}_t - \boldsymbol{\delta}_{t+1}}{t-k}\right\|^2$$
$$= \mathbb{E}\left\|\frac{\sum_{j=k+1}^t \left(\beta_1^{j-k} \tilde{\boldsymbol{m}}_k + (1-\beta_1)\sum_{h=k}^{j-1} \beta_1^{j-h-1} \boldsymbol{g}_h\right) + \boldsymbol{\delta}_t - \boldsymbol{\delta}_{t+1}}{t-k}\right\|^2$$

$$=\mathbb{E}\left\|\frac{1}{t-k}\sum_{j=k+1}^{t}\beta_1^{j-k}\tilde{\boldsymbol{m}}_k+\frac{1-\beta_1}{t-k}\sum_{j=k+1}^{t}\sum_{h=k}^{j-1}\beta_1^{j-h-1}\boldsymbol{g}_h+(\boldsymbol{\delta}_t-\boldsymbol{\delta}_{t+1})\right\|^2$$

$$\overset{\forall\eta>0}{\leq}(1+\eta)\mathbb{E}\left\|\frac{1}{t-k}\sum_{j=k+1}^{t}\beta_1^{j-k}\tilde{\boldsymbol{m}}_k\right\|^2+(1+1/\eta)\mathbb{E}\left\|\frac{1-\beta_1}{t-k}\sum_{j=k+1}^{t}\sum_{h=k}^{j-1}\beta_1^{j-h-1}\boldsymbol{g}_h+(\boldsymbol{\delta}_t-\boldsymbol{\delta}_{t+1})\right\|^2$$

$$\leq\frac{1+\eta}{t-k}\sum_{j=k+1}^{t}\mathbb{E}\left\|\beta_1^{j-k}\tilde{\boldsymbol{m}}_k\right\|^2+\frac{3(1+1/\eta)(1-\beta_1)}{t-k}\sum_{j=k+1}^{t}\sum_{h=k}^{j-1}\beta_1^{j-h-1}\boldsymbol{g}_h\mathbb{E}\|\boldsymbol{g}_h\|^2$$

$$+3(1+1/\eta)\mathbb{E}\|\boldsymbol{\delta}_t\|^2+3(1+1/\eta)\mathbb{E}\|\boldsymbol{\delta}_{t+1}\|^2$$

$$\overset{\eta=1/\beta_1-1}{\leq}(1+\eta)\beta_1^2\cdot\frac{3G_\infty^2d+24\Delta^2}{(1-\beta_1)^2}+3(1+1/\eta)G_\infty^2d+24(1+1/\eta)\Delta^2$$

$$=\beta_1\cdot\frac{3G_\infty^2d+24\Delta^2}{(1-\beta_1)^2}+\frac{3G_\infty^2d+24\Delta^2}{(1-\beta_1)^2}$$

$$=\frac{3G_\infty^2d+24\Delta^2}{(1-\beta_1)^2}.$$

On the other hand, if $t\notin\mathcal{T}_{\boldsymbol{u}}$, then

$$\mathbb{E}\left\|\boldsymbol{m}_{t+1}^{(i)}\right\|^2=\mathbb{E}\left\|\boldsymbol{m}_{t+\frac{1}{2}}^{(i)}\right\|^2=\mathbb{E}\left\|\beta_1\boldsymbol{m}_t^{(i)}+(1-\beta_1)\boldsymbol{g}_t^{(i)}\right\|^2$$

$$\leq\beta_1\mathbb{E}\left\|\beta_1\boldsymbol{m}_t^{(i)}\right\|^2+(1-\beta_1)\mathbb{E}\left\|\boldsymbol{g}_t^{(i)}\right\|^2\leq\frac{3G_\infty^2d+24\Delta^2}{(1-\beta_1)^2}.$$

For all the $t+\frac{1}{2}$ case, the inequality holds trivially due to Jensen Inequality. Finally, all the $\tilde{\cdot}$ bound can also be obtained via Jensen Inequality. And that completes the proof. $\qquad\square$

**Lemma 9.** *In Algorithm 1, for all the $t$ such that $t\notin\mathcal{T}_{\boldsymbol{v}}$, it holds that if we set learning rate*

$$\gamma\leq\min\left\{\frac{\beta_2^m}{4V_1L\sqrt{G_\infty^2+\epsilon}},\frac{2\sqrt{G_\infty^2+\epsilon}}{L}\right\},$$

*then,*

$$\sum_{t\notin\mathcal{T}_{\boldsymbol{v}}}\frac{\gamma\mathbb{E}\|\nabla f(\tilde{\boldsymbol{x}}_t)\|^2}{4\sqrt{G_\infty^2+\epsilon}}$$

$$\leq\sum_{t\notin\mathcal{T}_{\boldsymbol{v}}}\mathbb{E}f(\tilde{\boldsymbol{y}}_t)-\mathbb{E}f(\tilde{\boldsymbol{y}}_{t+1})+\frac{36\gamma^3H^2V_1(3G_\infty^2d+25\Delta^2)L^2(1+L)(G_\infty^2+\epsilon+1)(T-m)}{\beta_2^m(1-\beta_1)^4\sqrt{G_\infty^2+\epsilon}}$$

$$+\frac{L\gamma^2V_1\sigma^2(T-m)}{n\beta_2^m}+\frac{48\gamma^3V_1(H+1)^2(3G_\infty^2d+24\Delta^2)\sqrt{G_\infty^2+\epsilon}(T-m)}{\beta_2^m(1-\beta_1)^4}.$$

*Proof.* Since when $t\notin\mathcal{T}_{\boldsymbol{v}}$, it can either belongs to $\mathcal{T}_{\boldsymbol{u}}$ or not. We first prove the case for $t\in\mathcal{T}_{\boldsymbol{u}}$. From the definition of the auxiliary sequence, we obtain,

$$\tilde{\boldsymbol{y}}_{t+1}-\tilde{\boldsymbol{y}}_t=\tilde{\boldsymbol{x}}_{t+1}-\tilde{\boldsymbol{x}}_t-\frac{\gamma}{1-\beta_1}\left(\frac{\tilde{\boldsymbol{m}}_{t+1}}{\sqrt{\boldsymbol{v}_{t+1}+\epsilon}}-\frac{\tilde{\boldsymbol{m}}_t}{\sqrt{\boldsymbol{v}_t+\epsilon}}\right)-\left(\frac{\gamma\boldsymbol{\delta}_{t+1}}{\sqrt{\boldsymbol{v}_{t+1}+\epsilon}}-\frac{\gamma\boldsymbol{\delta}_t}{\sqrt{\boldsymbol{v}_t+\epsilon}}\right)$$

$$=\tilde{\boldsymbol{x}}_{t+1}-\tilde{\boldsymbol{x}}_t-\frac{\gamma}{(1-\beta_1)\sqrt{\boldsymbol{v}_t+\epsilon}}(\tilde{\boldsymbol{m}}_{t+1}-\tilde{\boldsymbol{m}}_t)-\frac{1}{\sqrt{\boldsymbol{v}_t+\epsilon}}(\gamma\boldsymbol{\delta}_{t+1}-\gamma\boldsymbol{\delta}_t)$$

$$=\tilde{\boldsymbol{x}}_{t+\frac{1}{2}}-\tilde{\boldsymbol{x}}_t-\frac{\gamma}{(1-\beta_1)\sqrt{\boldsymbol{v}_t+\epsilon}}\left(\tilde{\boldsymbol{m}}_{t+\frac{1}{2}}-\tilde{\boldsymbol{m}}_t\right)$$

$$\underbrace{+\tilde{\boldsymbol{x}}_{t+1}-\tilde{\boldsymbol{x}}_{t+\frac{1}{2}}-\frac{\gamma}{(1-\beta_1)\sqrt{\boldsymbol{v}_t+\epsilon}}\left(\tilde{\boldsymbol{m}}_{t+1}-\tilde{\boldsymbol{m}}_{t+\frac{1}{2}}\right)-\frac{1}{\sqrt{\boldsymbol{v}_t+\epsilon}}(\gamma\boldsymbol{\delta}_{t+1}-\gamma\boldsymbol{\delta}_t)}_{=\boldsymbol{q}_t}$$

$$=-\frac{\gamma\tilde{\boldsymbol{m}}_t}{\sqrt{\boldsymbol{v}_t+\epsilon}}-\frac{\gamma}{(1-\beta_1)\sqrt{\boldsymbol{v}_t+\epsilon}}(\beta_1\tilde{\boldsymbol{m}}_t+(1-\beta_1)\tilde{\boldsymbol{g}}_t-\tilde{\boldsymbol{m}}_t)+\boldsymbol{q}_t$$

$$= -\frac{\gamma \tilde{\boldsymbol{g}}_t}{\sqrt{\boldsymbol{v}+\epsilon}} + \boldsymbol{q}_t.$$

From Assumption 1, we have

$$\mathbb{E}f(\tilde{\boldsymbol{y}}_{t+1}) - \mathbb{E}f(\tilde{\boldsymbol{y}}_t) \leq \mathbb{E}\langle \nabla f(\tilde{\boldsymbol{y}}_t), \tilde{\boldsymbol{y}}_{t+1} - \tilde{\boldsymbol{y}}_t \rangle + \frac{L}{2}\mathbb{E}\|\tilde{\boldsymbol{y}}_{t+1} - \tilde{\boldsymbol{y}}_t\|^2$$

$$= \underbrace{-\gamma\mathbb{E}\left\langle \nabla f(\tilde{\boldsymbol{y}}_t), \frac{\tilde{\boldsymbol{g}}_t}{\sqrt{\boldsymbol{v}_t+\epsilon}}\right\rangle}_{A_1} + \underbrace{L\gamma^2\mathbb{E}\left\|\frac{\tilde{\boldsymbol{g}}_t}{\sqrt{\boldsymbol{v}_t+\epsilon}}\right\|^2}_{A_2} \underbrace{-\gamma\mathbb{E}\langle \nabla f(\tilde{\boldsymbol{y}}_t), \boldsymbol{q}_t\rangle}_{A_3} + \underbrace{L\gamma^2\mathbb{E}\|\boldsymbol{q}_t\|^2}_{A_4}.$$

We now bound $A_1$ to $A_4$ separately. Note that from Lemma 8, the momentum term can be uniformly bounded by a constant. For brevity of the derivation, we use $M$ to denote such constant bound, and fit in its value at the end of the proof.

For $A_1$,

$$A_1$$
$$= -\gamma\mathbb{E}\left\langle \nabla f(\tilde{\boldsymbol{y}}_t), \frac{\tilde{\boldsymbol{g}}_t}{\sqrt{\boldsymbol{v}_t+\epsilon}}\right\rangle$$

$$= -\gamma\mathbb{E}\left\langle \nabla f(\tilde{\boldsymbol{y}}_t), \frac{\frac{1}{n}\sum_{i=1}^n \nabla f\left(\boldsymbol{x}_t^{(i)}\right)}{\sqrt{\boldsymbol{v}_t+\epsilon}}\right\rangle$$

$$= -\gamma\mathbb{E}\left\langle \nabla f(\tilde{\boldsymbol{x}}_t), \frac{\nabla f(\tilde{\boldsymbol{x}}_t)}{\sqrt{\boldsymbol{v}_t+\epsilon}}\right\rangle - \gamma\mathbb{E}\left\langle \nabla f(\tilde{\boldsymbol{x}}_t), \frac{\frac{1}{n}\sum_{i=1}^n \nabla f\left(\boldsymbol{x}_t^{(i)}\right) - \nabla f(\tilde{\boldsymbol{x}}_t)}{\sqrt{\boldsymbol{v}_t+\epsilon}}\right\rangle$$

$$\quad - \gamma\mathbb{E}\left\langle \nabla f(\tilde{\boldsymbol{y}}_t) - \nabla f(\tilde{\boldsymbol{x}}_t), \frac{\nabla f(\tilde{\boldsymbol{x}}_t)}{\sqrt{\boldsymbol{v}_t+\epsilon}}\right\rangle - \gamma\mathbb{E}\left\langle \nabla f(\tilde{\boldsymbol{y}}_t) - \nabla f(\tilde{\boldsymbol{x}}_t), \frac{\frac{1}{n}\sum_{i=1}^n \nabla f\left(\boldsymbol{x}_t^{(i)}\right) - \nabla f(\tilde{\boldsymbol{x}}_t)}{\sqrt{\boldsymbol{v}_t+\epsilon}}\right\rangle$$

$$\leq -\frac{\gamma\mathbb{E}\|\nabla f(\tilde{\boldsymbol{x}}_t)\|^2}{\sqrt{G_\infty^2+\epsilon}} + \frac{\gamma\eta_1}{2}\mathbb{E}\|\nabla f(\tilde{\boldsymbol{x}}_t)\|^2 + \frac{\gamma}{2\eta_1}\mathbb{E}\left\|\frac{\frac{1}{n}\sum_{i=1}^n \nabla f\left(\boldsymbol{x}_t^{(i)}\right) - \nabla f(\tilde{\boldsymbol{x}}_t)}{\sqrt{\boldsymbol{v}_t+\epsilon}}\right\|^2 + \frac{\gamma\eta_1}{2}\mathbb{E}\|\nabla f(\tilde{\boldsymbol{x}}_t)\|^2$$

$$\quad + \frac{\gamma}{2\eta_1}\mathbb{E}\left\|\frac{\nabla f(\tilde{\boldsymbol{y}}_t) - \nabla f(\tilde{\boldsymbol{x}}_t)}{\sqrt{\boldsymbol{v}_t+\epsilon}}\right\|^2 + \frac{\gamma\eta_1}{2}\mathbb{E}\|\nabla f(\tilde{\boldsymbol{y}}_t) - \nabla f(\tilde{\boldsymbol{x}}_t)\|^2 + \frac{\gamma}{2\eta_1}\mathbb{E}\left\|\frac{\frac{1}{n}\sum_{i=1}^n \nabla f\left(\boldsymbol{x}_t^{(i)}\right) - \nabla f(\tilde{\boldsymbol{x}}_t)}{\sqrt{\boldsymbol{v}_t+\epsilon}}\right\|^2$$

$$\leq -\left(\frac{\gamma}{\sqrt{G_\infty^2+\epsilon}} - \gamma\eta_1\right)\mathbb{E}\|\nabla f(\tilde{\boldsymbol{x}}_t)\|^2 + \frac{\gamma V_1 L^2}{\beta_2^m \eta_1 n}\sum_{i=1}^n \mathbb{E}\left\|\boldsymbol{x}_t^{(i)} - \tilde{\boldsymbol{x}}_t\right\|^2 + \left(\frac{\gamma V_1 L^2}{2\beta_2^m \eta_1} + \frac{\gamma\eta_1 L^2}{2}\right)\mathbb{E}\|\tilde{\boldsymbol{y}}_t - \tilde{\boldsymbol{x}}_t\|^2,$$

where in the last step we use Assumption 1, Lemma 2 and Lemma 4. For the second term, denote the last sync step before $t$ is $k$, then we have:

$$\mathbb{E}\left\|\boldsymbol{x}_t^{(i)} - \tilde{\boldsymbol{x}}_t\right\|^2 = \mathbb{E}\left\|\boldsymbol{x}_t^{(i)} - \boldsymbol{x}_k^{(i)} - (\tilde{\boldsymbol{x}}_t - \tilde{\boldsymbol{x}}_k)\right\|^2$$

$$\leq 2\mathbb{E}\left\|\boldsymbol{x}_t^{(i)} - \boldsymbol{x}_k^{(i)}\right\|^2 + 2\mathbb{E}\|\tilde{\boldsymbol{x}}_t - \tilde{\boldsymbol{x}}_k\|^2$$

$$\leq 2\gamma^2\mathbb{E}\left\|\sum_{j=k}^{t-1}\frac{\boldsymbol{m}_j^{(i)}}{\sqrt{\boldsymbol{v}_t+\epsilon}}\right\|^2 + 2\gamma^2\mathbb{E}\left\|\frac{1}{n}\sum_{i=1}^n\sum_{j=k}^{t-1}\frac{\boldsymbol{m}_j^{(i)}}{\sqrt{\boldsymbol{v}_t+\epsilon}}\right\|^2 \quad (5)$$

$$\leq 2\gamma^2(t-k)\sum_{j=k}^{t-1}\mathbb{E}\left\|\frac{\boldsymbol{m}_j^{(i)}}{\sqrt{\boldsymbol{v}_t+\epsilon}}\right\|^2 + 2\gamma^2(t-k)\frac{1}{n}\sum_{i=1}^n\sum_{j=k}^{t-1}\mathbb{E}\left\|\frac{\boldsymbol{m}_j^{(i)}}{\sqrt{\boldsymbol{v}_t+\epsilon}}\right\|^2$$

$$\leq \frac{4\gamma^2 H^2 V_1 M}{\beta_2^m},$$

where the first step holds because Lemma 4, Lemma 2, and the fact that at the sync step $k$, $\tilde{\boldsymbol{x}}_k = \boldsymbol{x}_k^{(i)}$. For the third term, we have

$$
\begin{aligned}
\mathbb{E}\|\tilde{\boldsymbol{y}}_t - \tilde{\boldsymbol{x}}_t\|^2 &= \mathbb{E}\left\|\frac{\gamma \tilde{\boldsymbol{m}}_t}{(1-\beta_1)\sqrt{\boldsymbol{v}_t+\epsilon}} + \frac{\gamma \boldsymbol{\delta}_t}{\sqrt{\boldsymbol{v}_t+\epsilon}}\right\|^2 \\
&\leq \frac{2\gamma^2 V_1}{\beta_2^m(1-\beta_1)^2}\mathbb{E}\|\tilde{\boldsymbol{m}}_t\|^2 + \frac{2V_1}{\beta_2^m}\mathbb{E}\|\gamma \boldsymbol{\delta}_t\|^2 \\
&\overset{Lemma\ 7}{\leq} \frac{2\gamma^2 V_1 M}{\beta_2^m(1-\beta_1)^2} + \frac{2\gamma^2 V_1}{\beta_2^m} \cdot 4\Delta^2 \\
&\leq \frac{2\gamma^2 V_1 M}{\beta_2^m(1-\beta_1)^2} + \frac{8\gamma^2 V_1 \Delta^2}{\beta_2^m},
\end{aligned}
\tag{6}
$$

where we again apply the Lemma 2 and Lemma 4. Then we can get

$$
\begin{aligned}
A_1 &\leq -\left(\frac{\gamma}{\sqrt{G_\infty^2+\epsilon}} - \gamma\eta_1\right)\mathbb{E}\|\nabla f(\tilde{\boldsymbol{x}}_t)\|^2 + \frac{\gamma V_1 L^2}{\beta_2^m \eta_1 n}\sum_{i=1}^n \mathbb{E}\left\|\boldsymbol{x}_t^{(i)} - \tilde{\boldsymbol{x}}_t\right\|^2 \\
&\quad + \left(\frac{\gamma V_1 L^2}{2\beta_2^m \eta_1} + \frac{\gamma\eta_1 L^2}{2}\right)\mathbb{E}\|\tilde{\boldsymbol{y}}_t - \tilde{\boldsymbol{x}}_t\|^2 \\
&\leq -\left(\frac{\gamma}{\sqrt{G_\infty^2+\epsilon}} - \gamma\eta_1\right)\mathbb{E}\|\nabla f(\tilde{\boldsymbol{x}}_t)\|^2 + \frac{4\gamma^3 H^2 V_1^2 L^2 M}{\beta_2^m \eta_1} \\
&\quad + \left(\frac{\gamma V_1 L^2}{2\beta_2^m \eta_1} + \frac{\gamma\eta_1 L^2}{2}\right) \cdot \left(\frac{2\gamma^2 V_1 M}{\beta_2^m(1-\beta_1)^2} + \frac{8\gamma^2 V_1 \Delta^2}{\beta_2^m}\right) \\
&\leq -\left(\frac{\gamma}{\sqrt{G_\infty^2+\epsilon}} - \gamma\eta_1\right)\mathbb{E}\|\nabla f(\tilde{\boldsymbol{x}}_t)\|^2 + \frac{4\gamma^3 H^2 V_1^2 L^2 M}{\beta_2^m \eta_1} + \frac{\gamma^3 V_1^2 M L^2}{\eta_1 \beta_2^{2m}(1-\beta_1)^2} \\
&\quad + \frac{\gamma^3 \eta_1 V_1 M L^2}{\beta_2^m(1-\beta_1)^2} + \frac{4\gamma^3 V_1^2 \Delta^2 L^2}{\eta_1 \beta_2^{2m}} + \frac{4\gamma^3 \eta_1 V_1 \Delta^2 L^2}{\beta_2^m}.
\end{aligned}
$$

where in the second step we reuse Equation (5). Next we can bound $A_2$ as follows

$$
\begin{aligned}
A_2 &= L\gamma^2 \mathbb{E}\left\|\frac{\tilde{\boldsymbol{g}}_t}{\sqrt{\boldsymbol{v}_t+\epsilon}}\right\|^2 \\
&\leq \frac{L\gamma^2 V_1}{\beta_2^m}\mathbb{E}\left\|\frac{1}{n}\sum_{i=1}^n \boldsymbol{g}_t^{(i)}\right\|^2 \\
&\leq \frac{L\gamma^2 V_1 \sigma^2}{n\beta_2^m} + \frac{L\gamma^2 V_1}{\beta_2^m}\mathbb{E}\left\|\frac{1}{n}\sum_{i=1}^n \nabla f\left(\boldsymbol{x}_t^{(i)}\right)\right\|^2 \\
&\leq \frac{L\gamma^2 V_1 \sigma^2}{n\beta_2^m} + \frac{2L\gamma^2 V_1}{\beta_2^m}\mathbb{E}\left\|\frac{1}{n}\sum_{i=1}^n \nabla f\left(\boldsymbol{x}_t^{(i)}\right) - \nabla f(\tilde{\boldsymbol{x}}_t)\right\|^2 + \frac{2L\gamma^2 V_1}{\beta_2^m}\mathbb{E}\|\nabla f(\tilde{\boldsymbol{x}}_t)\|^2 \\
&\leq \frac{L\gamma^2 V_1 \sigma^2}{n\beta_2^m} + \frac{2L\gamma^2 V_1 L^2}{n\beta_2^m}\sum_{i=1}^n \mathbb{E}\left\|\boldsymbol{x}_t^{(i)} - \tilde{\boldsymbol{x}}_t\right\|^2 + \frac{2L\gamma^2 V_1}{\beta_2^m}\mathbb{E}\|\nabla f(\tilde{\boldsymbol{x}}_t)\|^2 \\
&\leq \frac{L\gamma^2 V_1 \sigma^2}{n\beta_2^m} + \frac{8\gamma^3 V_1^2 H^2 M L^3}{\beta_2^m} + \frac{2L\gamma^2 V_1}{\beta_2^m}\mathbb{E}\|\nabla f(\tilde{\boldsymbol{x}}_t)\|^2,
\end{aligned}
$$

where in the sixth step we reuse Equation (5). For $A_3$,

$$
\begin{aligned}
A_3 &= -\gamma\mathbb{E}\langle\nabla f(\tilde{\boldsymbol{y}}_t), \boldsymbol{q}_t\rangle \\
&= -\gamma\mathbb{E}\langle\nabla f(\tilde{\boldsymbol{x}}_t), \boldsymbol{q}_t\rangle - \gamma\mathbb{E}\langle\nabla f(\tilde{\boldsymbol{y}}_t) - \nabla f(\tilde{\boldsymbol{x}}_t), \boldsymbol{q}_t\rangle \\
&\overset{\forall \eta_2 > 0}{\leq} \frac{\gamma\eta_2}{2}\mathbb{E}\|\nabla f(\tilde{\boldsymbol{x}}_t)\|^2 + \frac{\gamma\eta_2}{2}\mathbb{E}\|\nabla f(\tilde{\boldsymbol{y}}_t) - \nabla f(\tilde{\boldsymbol{x}}_t)\|^2 + \frac{\gamma}{\eta_2}\mathbb{E}\|\boldsymbol{q}_t\|^2 \\
&\leq \frac{\gamma\eta_2}{2}\mathbb{E}\|\nabla f(\tilde{\boldsymbol{x}}_t)\|^2 + \frac{\gamma\eta_2 L^2}{2} \cdot \left(\frac{2\gamma^2 V_1 M}{\beta_2^m(1-\beta_1)^2} + \frac{8\gamma^2 V_1 \Delta^2}{\beta_2^m}\right) + \frac{\gamma}{\eta_2}\mathbb{E}\|\boldsymbol{q}_t\|^2
\end{aligned}
$$

$$\leq \frac{\gamma\eta_2}{2}\mathbb{E}\|\nabla f(\tilde{\boldsymbol{x}}_t)\|^2 + \frac{\gamma^3\eta_2 V_1 M L^2}{\beta_2^m(1-\beta_1)^2} + \frac{4\gamma^3\eta_2 V_1 \Delta^2 L^2}{\beta_2^m} + \frac{\gamma}{\eta_2}\mathbb{E}\|\boldsymbol{q}_t\|^2,$$

where in the last step we reuse Equation (6). Combine the bound of $A_1$ to $A_4$, we obtain

$$\mathbb{E}f(\tilde{\boldsymbol{y}}_{t+1}) - \mathbb{E}f(\tilde{\boldsymbol{y}}_t)$$

$$\leq -\left(\frac{\gamma}{\sqrt{G_\infty^2+\epsilon}} - \gamma\eta_1 - \frac{\gamma\eta_2}{2}\right)\mathbb{E}\|\nabla f(\tilde{\boldsymbol{x}}_t)\|^2 + \frac{4\gamma^3 H^2 V_1^2 L^2 M}{\beta_2^m \eta_1} + \frac{\gamma^3 V_1^2 M L^2}{\eta_1 \beta_2^{2m}(1-\beta_1)^2}$$

$$+ \frac{\gamma^3\eta_1 V_1 M L^2}{\beta_2^m(1-\beta_1)^2} + \frac{4\gamma^3 V_1^2 \Delta^2 L^2}{\eta_1 \beta_2^{2m}}$$

$$+ \frac{4\gamma^3\eta_1 V_1 \Delta^2 L^2}{\beta_2^m} + \frac{L\gamma^2 V_1 \sigma^2}{n\beta_2^m} + \frac{8\gamma^3 V_1^2 H^2 M L^3}{\beta_2^m} + \frac{2L\gamma^2 V_1}{\beta_2^m}\mathbb{E}\|\nabla f(\tilde{\boldsymbol{x}}_t)\|^2$$

$$+ \frac{\gamma^3\eta_2 V_1 M L^2}{\beta_2^m(1-\beta_1)^2} + \frac{4\gamma^3\eta_2 V_1 \Delta^2 L^2}{\beta_2^m} + \left(\frac{\gamma}{\eta_2} + L\gamma^2\right)\mathbb{E}\|\boldsymbol{q}_t\|^2.$$

We set the two constants $\eta_1,\eta_2$ as

$$\eta_1 = \frac{1}{4\sqrt{G_\infty^2+\epsilon}}$$

$$\eta_2 = \frac{1}{2\sqrt{G_\infty^2+\epsilon}},$$

then we have,

$$\mathbb{E}f(\tilde{\boldsymbol{y}}_{t+1}) - \mathbb{E}f(\tilde{\boldsymbol{y}}_t)$$

$$\leq -\left(\frac{\gamma}{\sqrt{G_\infty^2+\epsilon}} - \gamma\eta_1 - \frac{\gamma\eta_2}{2}\right)\mathbb{E}\|\nabla f(\tilde{\boldsymbol{x}}_t)\|^2 + \frac{4\gamma^3 H^2 V_1^2 L^2 M}{\beta_2^m \eta_1} + \frac{\gamma^3 V_1^2 M L^2}{\eta_1 \beta_2^{2m}(1-\beta_1)^2}$$

$$+ \frac{\gamma^3\eta_1 V_1 M L^2}{\beta_2^m(1-\beta_1)^2} + \frac{4\gamma^3 V_1^2 \Delta^2 L^2}{\eta_1 \beta_2^{2m}}$$

$$+ \frac{4\gamma^3\eta_1 V_1 \Delta^2 L^2}{\beta_2^m} + \frac{L\gamma^2 V_1 \sigma^2}{n\beta_2^m} + \frac{8\gamma^3 V_1^2 H^2 M L^3}{\beta_2^m} + \frac{2L\gamma^2 V_1}{\beta_2^m}\mathbb{E}\|\nabla f(\tilde{\boldsymbol{x}}_t)\|^2$$

$$+ \frac{\gamma^3\eta_2 V_1 M L^2}{\beta_2^m(1-\beta_1)^2} + \frac{4\gamma^3\eta_2 V_1 \Delta^2 L^2}{\beta_2^m} + \left(\frac{\gamma}{\eta_2} + L\gamma^2\right)\mathbb{E}\|\boldsymbol{q}_t\|^2$$

$$\leq -\left(\frac{\gamma}{2\sqrt{G_\infty^2+\epsilon}} - \frac{2L\gamma^2 V_1}{\beta_2^m}\right)\mathbb{E}\|\nabla f(\tilde{\boldsymbol{x}}_t)\|^2 + \frac{36\gamma^3 H^2 V_1(M+\Delta^2)L^2(1+L)(G_\infty^2+\epsilon+1)}{\beta_2^m(1-\beta_1)^2\sqrt{G_\infty^2+\epsilon}}$$

$$+ \frac{L\gamma^2 V_1 \sigma^2}{n\beta_2^m} + \left(2\gamma\sqrt{G_\infty^2+\epsilon} + L\gamma^2\right)\mathbb{E}\|\boldsymbol{q}_t\|^2.$$

Finally, we need to bound the norm of $\boldsymbol{q}_t$. If we denote the last sync step was $k$ steps before $t$, then,

$$\boldsymbol{q}_t = \tilde{\boldsymbol{x}}_{t+1} - \tilde{\boldsymbol{x}}_{t+\frac{1}{2}} - \frac{\gamma}{(1-\beta_1)\sqrt{\boldsymbol{v}_t+\epsilon}}\left(\tilde{\boldsymbol{m}}_{t+1} - \tilde{\boldsymbol{m}}_{t+\frac{1}{2}}\right) - \frac{\gamma\boldsymbol{\delta}_{t+1} - \gamma\boldsymbol{\delta}_t}{\sqrt{\boldsymbol{v}_t+\epsilon}}$$

$$= \tilde{\boldsymbol{x}}_{t+1} - \tilde{\boldsymbol{x}}_{t-k+1} + \tilde{\boldsymbol{x}}_{t-k+1} - \tilde{\boldsymbol{x}}_{t+\frac{1}{2}} - \frac{\gamma}{(1-\beta_1)\sqrt{\boldsymbol{v}_t+\epsilon}}\left(\tilde{\boldsymbol{m}}_{t+1} - \tilde{\boldsymbol{m}}_{t+\frac{1}{2}}\right) - \frac{\gamma\boldsymbol{\delta}_{t+1} - \gamma\boldsymbol{\delta}_t}{\sqrt{\boldsymbol{v}_t+\epsilon}}$$

$$= -\frac{\gamma\tilde{\boldsymbol{u}}_{t+\frac{1}{2}}}{\sqrt{\boldsymbol{v}_t+\epsilon}} - \left(\sum_{j=t-k+1}^{t}\frac{\gamma\tilde{\boldsymbol{m}}_j}{\sqrt{\boldsymbol{v}_t+\epsilon}}\right) - \frac{\gamma\left(\tilde{\boldsymbol{m}}_{t+1} - \tilde{\boldsymbol{m}}_{t+\frac{1}{2}}\right)}{(1-\beta_1)\sqrt{\boldsymbol{v}_t+\epsilon}}$$

$$= -\frac{\gamma}{(1-\beta_1)\sqrt{\boldsymbol{v}_t+\epsilon}}\left(\tilde{\boldsymbol{m}}_{t+1} - \tilde{\boldsymbol{m}}_{t+\frac{1}{2}} + 2(1-\beta_1)\sum_{j=t-k+1}^{t}\tilde{\boldsymbol{m}}_j\right),$$

based on which we obtain

$$\mathbb{E}\|\boldsymbol{q}_t\|^2 = \mathbb{E}\left\|\frac{\gamma}{(1-\beta_1)\sqrt{\boldsymbol{v}_t+\epsilon}}\left(\tilde{\boldsymbol{m}}_{t+1} - \tilde{\boldsymbol{m}}_{t+\frac{1}{2}} + 2(1-\beta_1)\sum_{j=t-k+1}^{t}\tilde{\boldsymbol{m}}_j\right)\right\|^2$$

$$\leq \frac{\gamma^2 V_1}{\beta_2^m(1-\beta_1)^2}\left(3\mathbb{E}\|\tilde{\boldsymbol{m}}_{t+1}\|^2 + 3\mathbb{E}\left\|\tilde{\boldsymbol{m}}_{t+\frac{1}{2}}\right\|^2 + 12(1-\beta_1)^2 k\sum_{j=t-k+1}^{t}\mathbb{E}\|\tilde{\boldsymbol{m}}_j\|^2\right)$$

$$\leq \frac{12\gamma^2 V_1(H+1)^2 M}{\beta_2^m(1-\beta_1)^2}.$$

Put everything together, and let $\gamma$ fulfills

$$\gamma \leq \min\left\{\frac{\beta_2^m}{4V_1 L\sqrt{G_\infty^2+\epsilon}}, \frac{2\sqrt{G_\infty^2+\epsilon}}{L}\right\},$$

we finally obtain

$$\mathbb{E}f(\tilde{\boldsymbol{y}}_{t+1})-\mathbb{E}f(\tilde{\boldsymbol{y}}_t)$$

$$\leq -\frac{\gamma\mathbb{E}\|\nabla f(\tilde{\boldsymbol{x}}_t)\|^2}{4\sqrt{G_\infty^2+\epsilon}} + \frac{36\gamma^3 H^2 V_1(M+\Delta^2)L^2(1+L)(G_\infty^2+\epsilon+1)}{\beta_2^m(1-\beta_1)^2\sqrt{G_\infty^2+\epsilon}} + \frac{L\gamma^2 V_1\sigma^2}{n\beta_2^m}$$

$$+\frac{48\gamma^3 V_1(H+1)^2 M\sqrt{G_\infty^2+\epsilon}}{\beta_2^m(1-\beta_1)^2}.$$

To this end, we have provided bound to all the sync steps $t$ with ($t\notin\mathcal{T}_{\boldsymbol{v}}$ and $t\in\mathcal{T}_{\boldsymbol{u}}$). For all the $t$ with ($t\notin\mathcal{T}_{\boldsymbol{v}}$ and $t\notin\mathcal{T}_{\boldsymbol{u}}$), they can be seen as a special case of $\boldsymbol{q}_t=\boldsymbol{0}$. Since $A_3+A_4>0$, this bound will continue to hold for them, so that to sum over all the $t$ with $t\notin\mathcal{T}_{\boldsymbol{v}}$, we obtain

$$\sum_{t\notin\mathcal{T}_{\boldsymbol{v}}}\frac{\gamma\mathbb{E}\|\nabla f(\tilde{\boldsymbol{x}}_t)\|^2}{4\sqrt{G_\infty^2+\epsilon}}$$

$$\leq \sum_{t\notin\mathcal{T}_{\boldsymbol{v}}}\mathbb{E}f(\tilde{\boldsymbol{y}}_t)-\mathbb{E}f(\tilde{\boldsymbol{y}}_{t+1}) + \frac{36\gamma^3 H^2 V_1(3G_\infty^2 d+25\Delta^2)L^2(1+L)(G_\infty^2+\epsilon+1)(T-m)}{\beta_2^m(1-\beta_1)^4\sqrt{G_\infty^2+\epsilon}}$$

$$+\frac{L\gamma^2 V_1\sigma^2(T-m)}{n\beta_2^m} + \frac{48\gamma^3 V_1(H+1)^2(3G_\infty^2 d+24\Delta^2)\sqrt{G_\infty^2+\epsilon}(T-m)}{\beta_2^m(1-\beta_1)^4},$$

where we replace $M$ with Lemma 8. That completes the proof. $\qquad\square$

**Lemma 10.** *In Algorithm 1, For all the $t\geq 0$ that fulfills $\boldsymbol{v}_t\neq\boldsymbol{v}_{t+1}$, i.e. $t\in\mathcal{T}_{\boldsymbol{v}}$, if the learning rate fulfills*

$$\gamma < \frac{1}{6}$$

*, the following bound holds*

$$\sum_{t\in\mathcal{T}_{\boldsymbol{v}}}\frac{\gamma\mathbb{E}\|\nabla f(\tilde{\boldsymbol{x}}_t)\|^2}{4\sqrt{G_\infty^2+\epsilon}}$$

$$\leq \sum_{t\in\mathcal{T}_{\boldsymbol{v}}}\mathbb{E}f(\tilde{\boldsymbol{y}}_t)-\mathbb{E}f(\tilde{\boldsymbol{y}}_{t+1}) + \frac{2\gamma\sigma^2 m}{nL} + \frac{106\gamma H^2 V_1(M+\Delta^2)mL}{\beta_2^m(1-\beta_1)^2} + \frac{\gamma\sigma^2 m}{4n\sqrt{G_\infty^2+\epsilon}} + \frac{\gamma G_\infty^2 dm}{4\sqrt{G_\infty^2+\epsilon}}.$$

*Proof.* From the definition of the auxiliary sequence, we obtain,

$$\tilde{\boldsymbol{y}}_{t+1}-\tilde{\boldsymbol{y}}_t = \tilde{\boldsymbol{x}}_{t+1}-\tilde{\boldsymbol{x}}_t - \frac{\gamma}{1-\beta_1}\left(\frac{\tilde{\boldsymbol{m}}_{t+1}}{\sqrt{\boldsymbol{v}_{t+1}+\epsilon}} - \frac{\tilde{\boldsymbol{m}}_t}{\sqrt{\boldsymbol{v}_t+\epsilon}}\right) - \left(\frac{\gamma\boldsymbol{\delta}_{t+1}}{\sqrt{\boldsymbol{v}_{t+1}+\epsilon}} - \frac{\gamma\boldsymbol{\delta}_t}{\sqrt{\boldsymbol{v}_t+\epsilon}}\right).$$

Based on Assumption 1,

$$\mathbb{E}f(\tilde{\boldsymbol{y}}_{t+1})-\mathbb{E}f(\tilde{\boldsymbol{y}}_t)$$

$$\leq \mathbb{E}\langle\nabla f(\tilde{\boldsymbol{y}}_t),\tilde{\boldsymbol{y}}_{t+1}-\tilde{\boldsymbol{y}}_t\rangle + \frac{L}{2}\mathbb{E}\|\tilde{\boldsymbol{y}}_{t+1}-\tilde{\boldsymbol{y}}_t\|^2$$

$$\overset{\gamma\eta<1}{\leq} \frac{\eta\gamma}{2L}\mathbb{E}\|\nabla f(\tilde{\boldsymbol{y}}_t)\|^2 + \frac{L}{\eta\gamma}\mathbb{E}\|\tilde{\boldsymbol{y}}_{t+1}-\tilde{\boldsymbol{y}}_t\|^2$$

$$\leq \frac{\eta\gamma}{L}\mathbb{E}\|\nabla f(\tilde{\boldsymbol{x}}_t)\|^2 + \eta\gamma L\mathbb{E}\|\tilde{\boldsymbol{y}}_t-\tilde{\boldsymbol{x}}_t\|^2 + \frac{L}{\eta\gamma}\mathbb{E}\|\tilde{\boldsymbol{y}}_{t+1}-\tilde{\boldsymbol{y}}_t\|^2$$

$$\leq \frac{2\eta\gamma}{L}\mathbb{E}\left\|\nabla f(\tilde{\boldsymbol{x}}_t)-\frac{1}{n}\sum_{i=1}^n\nabla f\left(\boldsymbol{x}_t^{(i)}\right)\right\|^2 + \frac{2\eta\gamma}{L}\mathbb{E}\left\|\frac{1}{n}\sum_{i=1}^n\nabla f\left(\boldsymbol{x}_t^{(i)}\right)-\frac{1}{n}\sum_{i=1}^n\boldsymbol{g}_t^{(i)}\right\|^2 + \eta\gamma L\mathbb{E}\|\tilde{\boldsymbol{y}}_t-\tilde{\boldsymbol{x}}_t\|^2$$

$$+\frac{L}{\eta\gamma}\mathbb{E}\|\tilde{\boldsymbol{y}}_{t+1}-\tilde{\boldsymbol{y}}_t\|^2$$

$$\leq \frac{2\eta\gamma L}{n}\sum_{i=1}^{n}\mathbb{E}\left\|\tilde{\boldsymbol{x}}_t-\boldsymbol{x}_t^{(i)}\right\|^2+\frac{2\eta\gamma\sigma^2}{nL}+\eta\gamma L\mathbb{E}\|\tilde{\boldsymbol{y}}_t-\tilde{\boldsymbol{x}}_t\|^2+\frac{L}{\eta\gamma}\mathbb{E}\|\tilde{\boldsymbol{y}}_{t+1}-\tilde{\boldsymbol{y}}_t\|^2.$$

We now bound the three norm terms separately. From Equation (5), we obtain for the first term,

$$\mathbb{E}\left\|\boldsymbol{x}_t^{(i)}-\tilde{\boldsymbol{x}}_t\right\|^2\leq\frac{4\gamma^2H^2V_1M}{\beta_2^m},$$

where we again use $M$ to denote the constant bound from Lemma 8 for brevity. On the other hand, based on a similar derivation to Equation (6), we obtain

$$\mathbb{E}\|\tilde{\boldsymbol{y}}_t-\tilde{\boldsymbol{x}}_t\|^2\leq\frac{2\gamma^2V_1M}{\beta_2^m(1-\beta_1)^2}+\frac{8\gamma^2V_1\Delta^2}{\beta_2^m}.$$

Finally, for the last norm, it's possible that the update towards $t+1$ step contains synchronization on the buffer. So that we need to discuss the two cases separately. First, for all the $t\in\mathcal{T}_{\boldsymbol{u}}$, denote the last sync step before $t$ is $k$, then we have

$$\mathbb{E}\|\tilde{\boldsymbol{y}}_{t+1}-\tilde{\boldsymbol{y}}_t\|^2$$

$$=\mathbb{E}\left\|\tilde{\boldsymbol{x}}_{t+1}-\tilde{\boldsymbol{x}}_t-\frac{\gamma}{1-\beta_1}\left(\frac{\tilde{\boldsymbol{m}}_{t+1}}{\sqrt{\boldsymbol{v}_{t+1}+\epsilon}}-\frac{\tilde{\boldsymbol{m}}_t}{\sqrt{\boldsymbol{v}_t+\epsilon}}\right)-\left(\frac{\gamma\boldsymbol{\delta}_{t+1}}{\sqrt{\boldsymbol{v}_{t+1}+\epsilon}}-\frac{\gamma\boldsymbol{\delta}_t}{\sqrt{\boldsymbol{v}_t+\epsilon}}\right)\right\|^2$$

$$\leq 7\mathbb{E}\left\|\tilde{\boldsymbol{x}}_{t+1}-\tilde{\boldsymbol{x}}_{t-k+1}\right\|^2+7\mathbb{E}\left\|\tilde{\boldsymbol{x}}_{t-k+1}-\tilde{\boldsymbol{x}}_{t+\frac{1}{2}}\right\|^2+7\mathbb{E}\left\|\tilde{\boldsymbol{x}}_{t+\frac{1}{2}}-\tilde{\boldsymbol{x}}_t\right\|^2+\frac{7\gamma^2}{1-\beta_1}\mathbb{E}\left\|\frac{\tilde{\boldsymbol{m}}_{t+1}}{\sqrt{\boldsymbol{v}_{t+1}+\epsilon}}\right\|^2$$

$$+\frac{7\gamma^2}{1-\beta_1}\mathbb{E}\left\|\frac{\tilde{\boldsymbol{m}}_t}{\sqrt{\boldsymbol{v}_t+\epsilon}}\right\|^2+7\mathbb{E}\left\|\frac{\gamma\boldsymbol{\delta}_{t+1}}{\sqrt{\boldsymbol{v}_{t+1}+\epsilon}}\right\|^2+7\mathbb{E}\left\|\frac{\gamma\boldsymbol{\delta}_t}{\sqrt{\boldsymbol{v}_t+\epsilon}}\right\|^2$$

$$\leq 7\mathbb{E}\left\|\tilde{\boldsymbol{x}}_{t+1}-\tilde{\boldsymbol{x}}_{t-k+1}\right\|^2+7\mathbb{E}\left\|\tilde{\boldsymbol{x}}_{t-k+1}-\tilde{\boldsymbol{x}}_{t+\frac{1}{2}}\right\|^2+7\gamma^2\mathbb{E}\left\|\frac{\tilde{\boldsymbol{m}}_t}{\sqrt{\boldsymbol{v}_t+\epsilon}}\right\|^2+\frac{7\gamma^2}{1-\beta_1}\mathbb{E}\left\|\frac{\tilde{\boldsymbol{m}}_{t+1}}{\sqrt{\boldsymbol{v}_{t+1}+\epsilon}}\right\|^2$$

$$+\frac{7\gamma^2}{1-\beta_1}\mathbb{E}\left\|\frac{\tilde{\boldsymbol{m}}_t}{\sqrt{\boldsymbol{v}_t+\epsilon}}\right\|^2+7\mathbb{E}\left\|\frac{\gamma\boldsymbol{\delta}_{t+1}}{\sqrt{\boldsymbol{v}_{t+1}+\epsilon}}\right\|^2+7\mathbb{E}\left\|\frac{\gamma\boldsymbol{\delta}_t}{\sqrt{\boldsymbol{v}_t+\epsilon}}\right\|^2$$

$$\leq 7\mathbb{E}\left\|\frac{\sum_{j=t-k+1}^{t}\gamma\tilde{\boldsymbol{m}}_j+\boldsymbol{\delta}_t-\boldsymbol{\delta}_{t+1}}{\sqrt{\boldsymbol{v}_k+\epsilon}}\right\|^2+7\mathbb{E}\left\|\frac{\sum_{j=t-k+1}^{t}\gamma\tilde{\boldsymbol{m}}_j}{\sqrt{\boldsymbol{v}_k+\epsilon}}\right\|^2+7\gamma^2\mathbb{E}\left\|\frac{\tilde{\boldsymbol{m}}_t}{\sqrt{\boldsymbol{v}_t+\epsilon}}\right\|^2$$

$$+\frac{7\gamma^2}{1-\beta_1}\mathbb{E}\left\|\frac{\tilde{\boldsymbol{m}}_{t+1}}{\sqrt{\boldsymbol{v}_{t+1}+\epsilon}}\right\|^2+\frac{7\gamma^2}{1-\beta_1}\mathbb{E}\left\|\frac{\tilde{\boldsymbol{m}}_t}{\sqrt{\boldsymbol{v}_t+\epsilon}}\right\|^2+7\mathbb{E}\left\|\frac{\gamma\boldsymbol{\delta}_{t+1}}{\sqrt{\boldsymbol{v}_{t+1}+\epsilon}}\right\|^2+7\mathbb{E}\left\|\frac{\gamma\boldsymbol{\delta}_t}{\sqrt{\boldsymbol{v}_t+\epsilon}}\right\|^2$$

$$\leq\frac{105\gamma^2H^2V_1(M+\Delta^2)}{\beta_2^m(1-\beta_1)^2},$$

where in the last step we use Lemma 7, 8 and 4. It is straightforward to verify that this bound also holds for $t\notin\mathcal{T}_{\boldsymbol{u}}$ (since there will be no noise from the sync step). Combine the three norm term bounds, we obtain

$$\mathbb{E}f(\tilde{\boldsymbol{y}}_{t+1})-\mathbb{E}f(\tilde{\boldsymbol{y}}_t)$$

$$\leq\frac{2\eta\gamma L}{n}\sum_{i=1}^{n}\mathbb{E}\left\|\tilde{\boldsymbol{x}}_t-\boldsymbol{x}_t^{(i)}\right\|^2+\frac{2\eta\gamma\sigma^2}{nL}+\eta\gamma L\mathbb{E}\|\tilde{\boldsymbol{y}}_t-\tilde{\boldsymbol{x}}_t\|^2+\frac{L}{\eta\gamma}\mathbb{E}\|\tilde{\boldsymbol{y}}_{t+1}-\tilde{\boldsymbol{y}}_t\|^2$$

$$=\frac{8\eta\gamma^3H^2V_1ML}{\beta_2^m}+\frac{2\eta\gamma\sigma^2}{nL}+\eta\gamma L\left(\frac{2\gamma^2V_1M}{\beta_2^m(1-\beta_1)^2}+\frac{8\gamma^2V_1\Delta^2}{\beta_2^m}\right)+\frac{105\gamma H^2V_1(M+\Delta^2)L}{\eta\beta_2^m(1-\beta_1)^2}$$

$$\leq\frac{2\eta\gamma\sigma^2}{nL}+\frac{18\eta\gamma^3H^2V_1ML}{\beta_2^m(1-\beta_1)^2}+\frac{105\gamma H^2V_1(M+\Delta^2)L}{\eta\beta_2^m(1-\beta_1)^2}$$

$$\leq\frac{2\gamma\sigma^2}{nL}+\frac{106\gamma H^2V_1(M+\Delta^2)L}{\beta_2^m(1-\beta_1)^2},$$

where in the last step we set $\eta=1$ and use the requirement that $\gamma<1/6$. Summing over all the $t\in\mathcal{T}_{\boldsymbol{v}}$, we get

$$0\leq\sum_{t\in\mathcal{T}_{\boldsymbol{v}}}\mathbb{E}f(\tilde{\boldsymbol{y}}_t)-\mathbb{E}f(\tilde{\boldsymbol{y}}_{t+1})+\frac{2\gamma\sigma^2m}{nL}+\frac{106\gamma H^2V_1(M+\Delta^2)mL}{\beta_2^m(1-\beta_1)^2}.$$

Adding $\frac{\gamma}{4\sqrt{G_\infty^2+\epsilon}}\sum_{t\in\mathcal{T}_v}\mathbb{E}\|\nabla f(\tilde{x}_t)\|^2$ on both sides, and note that

$$\sum_{t\in\mathcal{T}_v}\mathbb{E}\|\nabla f(\tilde{x}_t)\|^2 = \sum_{t\in\mathcal{T}_v}\mathbb{E}\|\nabla f(\tilde{x}_t)-\tilde{g}_t\|^2 + \sum_{t\in\mathcal{T}_v}\mathbb{E}\|\tilde{g}_t\|^2$$

$$\leq \frac{\sigma^2 m}{n} + G_\infty^2 dm.$$

We finally obtain

$$\sum_{t\in\mathcal{T}_v}\frac{\gamma\mathbb{E}\|\nabla f(\tilde{x}_t)\|^2}{4\sqrt{G_\infty^2+\epsilon}}$$

$$\leq \sum_{t\in\mathcal{T}_v}\mathbb{E}f(\tilde{y}_t)-\mathbb{E}f(\tilde{y}_{t+1})+\frac{2\gamma\sigma^2 m}{nL}+\frac{106\gamma H^2 V_1(M+\Delta^2)mL}{\beta_2^m(1-\beta_1)^2}+\frac{\gamma\sigma^2 m}{4n\sqrt{G_\infty^2+\epsilon}}+\frac{\gamma G_\infty^2 dm}{4\sqrt{G_\infty^2+\epsilon}}.$$

That completes the proof. $\square$

