# OpenReview forum: "Maximizing Communication Efficiency for Large-scale Training via 0/1 Adam"
_ICLR.cc/2023/Conference — ICLR 2023 poster_

### Official Review · Reviewer_SjhX · 2022-10-22

**Confidence:** 4
**Correctness:** 3
**Technical Novelty And Significance:** 3
**Empirical Novelty And Significance:** 3
**Recommendation:** 6

**Clarity, Quality, Novelty And Reproducibility:**

The paper is written well. The experimental setup and parameter settings seem to be decent and explained thoroughly. I liked the discussion about policy, convergence speed, and quality analysis. Novelty is not impressive but decent and impactful enough. With the given implementation and parameter details, it seems that there is no reproducibility issue.

**Strength And Weaknesses:**

They provide convergence guarantees with theoretical analysis and experimental results comparing 1-bit Adam on language modeling training and image-net training showing a much faster reduction in training loss in both cases. The authors also open-sourced their algorithm.

One missing piece in this setup is the generalization, I think. I believe the experimental setup might be decent, and the problems are large enough to demonstrate efficacy. But I am not sure about the broader applicability to various other applications of AI, such as siamese networks (wide-and-deep models) or GANs, where the scale might come from a different angle than language models.

**Summary Of The Paper:**

With the deep-learning revolution, large-scale training has been at the forefront of innovation. As the data sizes grow, the training time and cost explode. There have been techniques such as compression and bit quantization to alleviate these problems. These techniques work well in linear information but not in non-linear information. The paper proposes a new method called 0/1 Adam, leveraging insights within updates and stationary via steps.


**Summary Of The Review:**

The paper focuses on an important problem to speed up model training. Though there are some minor limitations on generalization, I think it is good enough to be represented in the venue. I wanted to vote 7/10 (but only 6 or 8 are available).

---

> ### Author Response · Authors · 2022-11-10
> **Response to Reviewer SjhX**
>
> We want to thank you for your positive reviews on our paper!
>
> Regarding your concern on the generalization: We have included additional evaluation on the CIFAR task (which is a different-scale task compared to BERT, GPT or ImageNet pretraining) in the revised paper Appendix C Figure 7. Given we have limited time to update paper during the rebuttal phase, we will include any incoming results in the final version.
>
> While we truly appreciate your useful suggestions on adding more tasks, we hope this does not let us overlook the main motivation of 0/1 Adam, which is the observation on the huge overhead in large-scale large language model pretraining. Compared to other models and tasks, pretraining these foundation models is most likely to incur overhead at large scale.

---

### Official Review · Reviewer_LNoo · 2022-10-23

**Confidence:** 2
**Correctness:** 3
**Technical Novelty And Significance:** 2
**Empirical Novelty And Significance:** 2
**Recommendation:** 5

**Clarity, Quality, Novelty And Reproducibility:**

Clarity:
Presentation of results has significant room for improvement and requires more explanation.  For instance, it is unclear exactly what is being demonstrated in Figure 5: is "w/o Skipping Round" equivalent to w/o Local Steps?  What are 1.2x/1.3x denoting?  Similarly, in Figure 2 (a-c), what is being denoted by 1.6x, 2x, and 1.4x?  One suggestion is to describe each figure and the corresponding results after they are presented.  As it stands, Figures 2-5 all appear before a single figures' results are described in the text.

Quality:
Writing quality can be significantly improved.  Furthermore, comparing against Adam in all results would greatly improve the paper.  E.g., "Since Tang et al. (2021) already demonstrated that 1-bit Adam has similar statistical results to Adam, we omit the comparison of end-to-end model accuracy to Adam for brevity." <- This is unreasonable; 1-bit Adam has been shown to provide near Adam accuracy, and so the comparison between the two may be ommitted.  However, the newly described algorithm (which is not equivalent to 1-bit Adam) has not been shown to achieve similar performance to Adam, and thus this must be demonstrated.  For the purposes of the paper, including Adam as a baseline in Figures 4 and 5 would not overcomplicate these results, but would significantly strengthen the paper.

Novelty:
0/1 Adam is an extension of the 1-bit Adam algorithm.  Additionally, much of the motivation and theoretical analysis strongly follow the 1-bit Adam paper.  The paper needs to do a better job of establishing that 0/1 Adam builds on this previous work, what exact theoretical results were presented in the 1-bit Adam algorithm and how the presented theoretical results stand as novel contributions, and exactly what are the novel contributions being made.  E.g., "We propose 0/1 Adam, a novel optimization method that addresses the non-linearity challenges in Adam when applying aggressive 1-bit quantization and local steps" <- disingenuous to call 0/1 Adam novel, but rather "0/1 Adam builds upon the 1-bit Adam algorithm."  Furthermore, the use of local SGD/steps for Adam has been previously addressed by the Cada algorithm of Chen et al.

Reproducibility:
The 1-bit Adam has been open-sourced in a deep learning library by the authors.  However, the presented experiments are non-trivial, and providing experimental code to reproduce the featured results would greatly help other researchers to further build on their work.

**Strength And Weaknesses:**

Strengths
-The paper extends 1-bit Adam in a natural and intuitive way.  Both adaptive variance-freezing and local SGD are promising directions to improve the original 1-bit Adam algorithm
-The results are compelling

Weaknesses
-Presentation of results could be significantly improved (see clarity section for further information).

-The writing and description of the novelty of the presented work requires extensive reevaluation.  The description of 0/1 Adam as a novel algorithm is not correct, it builds upon 1-bit Adam (also borrowing a large amount the analysis from the corresponding paper) and this point should be made clear throughout the paper (however, adaptive variance freezing is an important and novel contribution to the 1-bit Adam algorithm). Furthermore, 0/1 Adam is not the first work to consider local SGD for distributed algorithm.

-Even after reading the paper, it is unclear how adaptive variance-freezing or local SGD (i.e., skipping rounds/local steps) may be set in practice.  Indeed, these two policies must be known \emph{before} running Algorithm~\ref{1}.

-Claims are made without substantiation or are misleading.  E.g.:

         1) "the change of variance over steps in Adam is generally smooth" <- Please substantiate this claim.  In the 1-bit Adam paper, they
              assumed this was the case and stated so explicitly.

         2) "While this paradigm avoids compressing non-linear information with a one-time frozen variance, the experimental results from
             (Tang et al., 2021) indicate the fullprecision stage still incurs non-trivial overhead." <- Overhead discussed in the paper was with
              regards to vanilla Adam.  Please remove this claim or provide additional evidence to substantiate it

         3) "Furthermore, 1-bit Adam is restricted in the scope of gradient compression, and cannot be trivially adapted when other
              techniques are used, such as local steps." <- In the 1-bit Adam paper, several variants were
              implemented and contrasted against, including compressing the gradient and compressing the momentum.  Why can it not be
              trivially adapted?  Furthermore, how to leverage local SGD with Adam is not an open problem, previous work in [1] has tackled
              this. 1-bit Adam could trivially be combined with CADA.  Please reassess, and support, your claim.

         4) "Besides, the empirical success of (Tang et al., 2021) was not substantiated on generative models (GPT-style models), for instance,
              175B GPT-3 Brown et al. (2020), 530B MT-NLG Smith et al. (2022), etc." <- This is disingenuous; 1-bit Adam was evaluated on
              GANs (generative model) as well large-scale models (BERT).  530B MT-NLG Smith et al. was also
              published after the 1-bit Adam paper, and neither GPT-3 no MT-NLG are evaluated using 0/1 Adam.  I would recommend revising
              this claim, as well as elaborating why BERT is not a significant enough benchmark that GPT model evaluations are required.

References:
[1] Chen, Tianyi, et al. "Cada: Communication-adaptive distributed adam."
International Conference on Artificial Intelligence and
Statistics. PMLR, 2021.

**Summary Of The Paper:**

The paper introduces 0/1 Adam, a variant of the 1-bit Adam algorithm.  While the 1-bit Adam algorithm leveraged a fixed burn-in period wherein vanilla Adam was run before the variance term stabilized and was frozen, 0/1 Adam proposes an adaptive variance-freezing policy.  Additionally, 0/1 Adam leverages local SGD wherein synchronicity rounds among workers are skipped.  Convergence results similar to 1-bit Adam are provided for 0/1 Adam, and experimental results showing the efficacy of the algorithm for Imagenet and BERT are provided.

**Summary Of The Review:**

The paper makes some natural extensions to the 1-bit Adam algorithm, while providing convergence guarantees for the resulting 0/1 Adam algorithm.  Furthermore, the results seem compelling.  However, the presentation of the paper's novelty and contributions requires serious reevaluation.  Also, the general presentation and writing has significant room for improvement, particularly discussion of the presented results.  Important key details regarding figures are lacking or insufficient to understand exactly what is being stated.  Finally, while the extensions made from 1-bit Adam to produce 0/1 Adam make intuitive sense, it is unclear how exactly these are (or can be made) adaptive in practice, particularly given their predefined values in Algorithm 1.

---

> ### Author Response · Authors · 2022-11-10
> **Response to Reviewer LNoo (Part I)**
>
> We want to thank you for the detailed and constructive reviews! We noticed that the majority of your concerns are on the wording / presentation of the paper. We strongly agree with you that these sentences could have been made clearer and we have revised the paper to incorporate them (details below), please let us know if the answers below address your concerns.
>
> > The description of 0/1 Adam as a novel algorithm is not correct, it builds upon 1-bit Adam (also borrowing a large amount the analysis from the corresponding paper) and this point should be made clear throughout the paper (however, adaptive variance freezing is an important and novel contribution to the 1-bit Adam algorithm). Furthermore, 0/1 Adam is not the first work to consider local SGD for distributed algorithms.
>
> Thank you for this valuable feedback! We completely agree with you that 0/1 Adam is designed upon, and motivated from 1-bit Adam. We have updated the wording in related sentences in the intro (please refer to the red text in the revision.). We truly appreciate your assessment on the novelty in adaptive variance freezing. Regarding the novelty in adapting local steps and comparison to other related algorithms, please refer to the answer of comparing with CADA below.
>
> > "Furthermore, 1-bit Adam is restricted in the scope of gradient compression, and cannot be trivially adapted when other techniques are used, such as local steps." <- In the 1-bit Adam paper, several variants were implemented and contrasted against, including compressing the gradient and compressing the momentum.  Why can it not be trivially adapted?  Furthermore, how to leverage local SGD with Adam is not an open problem, previous work in [1] has tackled this. 1-bit Adam could trivially be combined with CADA.  Please reassess, and support, your claim.
>
> We thank you for sharing the related work of CADA with us! We would like to clarify that while CADA is a great work, it is not suitable for large model training which 1-bit Adam and 0/1 Adam are trying to address.
>
> First, CADA computes 2X more gradients compared to the original Adam (as suggested in their Algorithm 1). While it is fine for small models, it is not affordable for large-scale language model pretraining. As our profiling results in Table 3 suggest, doubling the computation could take even longer time than we’d save from communication. Note that the 2X gradient computation has not included other additional operations in CADA like norm computations, which also  takes non-trivial time on large high-dimensional  models.
>
> Second, CADA needs at least three additional buffers (of model size!) in the implementation (as shown in their Algorithm 1 and Equation (7)&(10)). Again, this is problematic for large models. Note that 1-bit Adam already requires two additional buffers of model size for storing the compression error; combining 1-bit Adam with CADA would incur non-trivial memory overhead.

---

> > ### Author Response · Authors · 2022-11-10
> > **Response to Reviewer LNoo (Part II)**
> >
> > > Even after reading the paper, it is unclear how adaptive variance-freezing or local SGD (i.e., skipping rounds/local steps) may be set in practice. Indeed, these two policies must be known before running the algorithm.
> >
> > Thank you for raising this question! On page 8 of the manuscript, we included a paragraph titled “Remarks on the selected policy” explaining how the policies are obtained by following the learning rate schedule, and so we do **not** need to “know” them in advance. To summarize: we only need to handpick an initial interval for communication before the training starts, and then the interval will adaptively increase/decrease following the magnitude of learning rate. This works well in practice and is justifiable from three aspects (**this clarification is included in the paper**): (1) It does not require too much hyperparameter tuning. Consider adopting a constant or decaying policy based on some utility function, capturing the dynamics of the training to reach a useful interval would require tedious searching on the hyperparameters and retraining; (2) Learning rate policy is well-motivated. As naturally, communication is a way of eliminating the difference among workers. If workers adopt a larger learning rate, more frequent communication will be needed since the workers are taking large steps in the weight space and so we require higher-frequent communication for them to reach consensus; (3) Learning rate schedule is universally used in all the ML/DL applications, and thus the method can be easily adapted to other applications. In fact, if we consider the scope of large model training, a typical learning rate schedule is a linear warm-up followed by a decaying phase, which is very similar to our test cases here.
> >
> > On the other hand, we want to elaborate that in practice, tuning hyperparameters can have different reasons, depending on the use cases: (1) to make an algorithm work properly (that is, the algorithm can easily fail without tuning); (2) to maximize the benefits from an algorithm. Note that the hyperparameter tuning in 0/1 Adam falls in (2). That’s because if we do no hyperparameter tuning and just naively set all the interval bound to 2, we can still obtain non-trivial 50% communication off.
> >
> > > "the change of variance over steps in Adam is generally smooth" <- Please substantiate this claim.
> >
> > Thank you for pointing this out! We acknowledge there could be confusion when using the term ‘smooth’. We want to clarify that the smoothness here does not stand for the term from optimization theory, but refers to the fact that the variance does not have abrupt change at a single step. This is justifiable both in theory and in practice: Empirically we profiled the changes of variance on a large-scale task (BERT-large) in Figure 1(a)(b) and demonstrated that the variance is not changing abruptly; theoretically note that $\beta_2$ in Adam is usually set to be very close to 1 (e.g. its default value is 0.999 in all ML frameworks). Considering in Adam
> >
> > new variance = $\beta_2$ * old variance + (1-$\beta_2$) * gradient square,
> >
> > this implies the new variance will stay close to its original value  in a single step. While this is an informal description, it suffices for us to leverage the insights there and build an intuition.
> >
> >
> > > "While this paradigm avoids compressing non-linear information with a one-time frozen variance, the experimental results from (Tang et al., 2021) indicate the full precision stage still incurs non-trivial overhead." <- Overhead discussed in the paper was with regards to vanilla Adam.  Please remove this claim or provide additional evidence to substantiate it.
> >
> > Thank you for pointing this out! In fact, we did include additional details in footnote 3 for this statement: “As illustrated in (Tang et al., 2021), when training BERT-Large on 64 GPUs using Ethernet, while the full-precision stage contains 15% of the total steps, it can take more than 50% of the entire training in terms of the wall-clock time. Concretely, it shows in (Tang et al., 2021) Section 7.1 that to train BERT-Large on 64 GPUs using Ethernet, the full-precision Adam takes 174.3 hours in total while 1-bit Adam takes 51.5 hours. By a simple calculation, we know that the full-precision stage of 1-bit Adam takes approximately 26.37 hours while the compression stage takes 25.13 hours.”

---

> > > ### Author Response · Authors · 2022-11-10
> > > **Response to LNoo (Part III)**
> > >
> > > > "Besides, the empirical success … 530B MT-NLG Smith et al. (2022), etc." <- This is disingenuous; 1-bit Adam was evaluated on GANs (generative model) as well large-scale models (BERT).  530B MT-NLG Smith et al. was also published after the 1-bit Adam paper, and neither GPT-3 no MT-NLG are evaluated using 0/1 Adam.
> > >
> > > We thank you for pointing out this confusion. We have revised the manuscript to change the wording of the generative model to only GPT-style model.
> > >
> > > > It is unclear exactly what is being demonstrated in Figure 5: is "w/o Skipping Round" equivalent to w/o Local Steps? What are 1.2x/1.3x denoting? Similarly, in Figure 2 (a-c), what is being denoted by 1.6x, 2x, and 1.4x? One suggestion is to describe each figure and the corresponding results after they are presented. As it stands, Figures 2-5 all appear before a single figures' results are described in the text.
> > >
> > > Thank you for raising this question! For Figure 5, we have updated the manuscript to change the term “skipping rounds” to “local steps”. The denotation of 1.2X/1.3X in Figure 5 and 1.6X/2X/1.4X in Figure 2 shows the speed up of 0/1 Adam over 1-bit Adam over the same number of samples. More concretely, it demonstrates that to process (learn with) the same number of samples in a task, 1-bit Adam takes 1.2X/1.3X/1.6X/2X/1.4X wall-clock time in that task compared to 0/1 Adam. We have updated the manuscript to make this clearer.
> > >
> > > > "Since Tang et al. (2021) already … for brevity." <- This is unreasonable; 1-bit Adam has been shown to provide near Adam accuracy, and so the comparison between the two may be omitted. However, the newly described algorithm (which is not equivalent to 1-bit Adam) has not been shown to achieve similar performance to Adam, and thus this must be demonstrated. For the purposes of the paper, including Adam as a baseline in Figures 4 and 5 would not overcomplicate these results, but would significantly strengthen the paper.
> > >
> > > Thank you for flagging this question! We agree with you that this should be made clearer. On the other hand, we did compare 0/1 Adam with original Adam both in terms of systematic throughput (Figure 3) and model accuracy (Table 1 & 2). In figure 4, since the original Adam uses much more bits than other algorithms (16 bits), we included it in the first place but found it makes it hard for us to compare other algorithms (plotting wise). And so we omit Adam there for brevity. For Figure 5, we are verifying a hypothesis that is invariant to Adam, which is skipping communication rounds in 1-bit Adam is actually important even if the amortized bits become close to 1.
> > >
> > > To make this clearer, we have updated Figure 4 with additional clarifications.
> > >
> > > > The paper needs to do a better job of establishing that 0/1 Adam builds on this previous work, what exact theoretical results were presented in the 1-bit Adam algorithm and how the presented theoretical results stand as novel contributions, and exactly what are the novel contributions being made. E.g., "We propose 0/1 Adam, a novel optimization method that addresses the non-linearity challenges in Adam when applying aggressive 1-bit quantization and local steps" <- disingenuous to call 0/1 Adam novel, but rather "0/1 Adam builds upon the 1-bit Adam algorithm."
> > >
> > > We thank you for raising this concern, we have updated the related sentences accordingly in the revision.

---

### Official Review · Reviewer_en2k · 2022-10-23

**Confidence:** 3
**Correctness:** 3
**Technical Novelty And Significance:** 3
**Empirical Novelty And Significance:** 3
**Recommendation:** 6

**Clarity, Quality, Novelty And Reproducibility:**

The clarity and quality of the paper are satisfying.  The method contains enough technical novelty to outperform existing work (1-bit Adam). The reproducibility should be good since the code is released and integrated into open-source repos.

**Strength And Weaknesses:**

Strength:
1. Firstly, the paper is generally well-written and easy to follow.
2. The evaluation is comprehensive. The authors provided experimental results on language model training (transfer learning and zero-shot evolution) and ImageNet classification, where the proposed optimizer can match the performance of fp16 Adam.
3. The proposed method achieves higher training throughputs across different benchmarks and hardware setups, which seems to be pretty general.
4. The motivation of the method is

Weakness:
1. The adjustment of hyper-parameter T_v and T_u seems a bit non-trivial (especially the latter one). Can the author confirms that the same hyper-parameter can scale across model scales without careful tuning? For example, does the same hyper-parameter scale to training on the CIFAR dataset with the same learning schedule?

**Summary Of The Paper:**

In this paper, the authors proposed 0/1 Adam to reduce communication during training. 0/1 Adam addresses the existing issues by adaptively freezing variance and linearly approximating momentum and parameter update locally. The authors provide theoretical convergence guarantee and experimental results on large-scale training tasks, showing that the proposed optimizer can achieve similar accuracy compared to fp16 training at a higher throughput.

**Summary Of The Review:**

Generally, I think this is a good paper. It contributes to the distributed training area by reducing communication to increase throughput. The method outperforms existing work by further reducing the high-precision training steps. My only concern is the difficulty of hyper-parameter adjustment.

---

> ### Author Response · Authors · 2022-11-10
> **Response to Reviewer en2k**
>
> Thank you for your positive feedback on our paper!
>
> Regarding your question on setting T_v and T_u (especially T_u): on page 8 of the manuscript, we included a paragraph titled “Remarks on the selected policy” explaining how we avoid redundant hyperparameter tuning by following the learning rate schedule. To summarize: we only need to handpick an initial interval for communication before the training starts, and then the interval will adaptively increase/decrease following the magnitude of learning rate. This works well in practice and is justifiable from three aspects (**this clarification is included in the paper**): (1) It does not require too much hyperparameter tuning. Consider adopting a constant or decaying policy based on some utility function, capturing the dynamics of the training to reach a useful interval would require tedious searching on the hyperparameters and retraining; (2) Learning rate policy is well-motivated. As naturally, communication is a way of eliminating the difference among workers. If workers adopt a larger learning rate, more frequent communication will be needed since the workers are taking large steps in the weight space and so we require higher-frequent communication for them to reach consensus; (3) Learning rate schedule is universally used in all the ML/DL applications, and thus the method can be easily adapted to other applications. In fact, if we consider the scope of large model training, a typical learning rate schedule is a linear warm-up followed by a decaying phase, which is very similar to our test cases here.
>
> On the other hand, we want to elaborate that in practice, tuning hyperparameters can have different reasons, depending on the use cases: (1) to make an algorithm work properly (that is, the algorithm can easily fail without tuning); (2) to maximize the benefits from an algorithm. Note that the hyperparameter tuning in 0/1 Adam falls in (2). That’s because if we do no hyperparameter tuning and just naively set all the interval bound to 2, we can still obtain non-trivial 50% communication off.
>
> In the meantime, we thank you for suggesting verifying the hyperparameter scales to other tasks like CIFAR. We have revised the paper and **included such results in Appendix C Figure 7**. These included results show that 0/1 Adam can easily scale to CIFAR dataset in terms of both training loss and validation accuracy with zero additional tuning.

---

### Author Response · Authors · 2022-11-12
**To all reviewers**

We thank all the reviewers for giving valuable and constructive feedback to our paper. To address the reviewers’ concerns, we have revised the paper to incorporate the following changes:

* (To Reviewer en2k, sjhX) We added additional experiments of running 0/1 Adam on the CIFAR datasets and confirmed the same hyperparameter scales to other tasks. Please refer to the revised manuscript Appendix C Figure 7 (in red text). Since there is limited time to update the paper during rebuttal, we will include the other incoming results in the final version of this paper.

* (To Reviewer LNoo) We fixed some presentation issues suggested by Reviewer LNoo. These include (all changes are highlighted with red text): rewording on the novelty description in the introduction; clarification on the wall-clock time in Figure 2; the “local steps” term in legend; clarification on Figure 4.

Please let us know if there is any additional concern or question, thank you!

Authors

---

### Decision · Program_Chairs · 2023-01-20

**Decision:**

Accept: poster

**Justification For Why Not Higher Score:**

It's a productive but incremental result on top of a popular paper. It's not earthshattering, and probably doesn't merit extra attention in the broader community.

**Justification For Why Not Lower Score:**

The paper is a productive improvement to one-bit adam that practitioners will probably find useful. I could see it being sufficiently incremental to merit rejection if absolutely needed.

**Metareview: Summary, Strengths And Weaknesses:**

**Summary:** This paper proposes a couple of improvements to one-bit Adam, which is a popular way to reduce the bandwidth requirements of training large language models. Specifically, the paper proposes (1) to freeze the variance over short intervals as opposed to one-bit adam's technique of freezing it permanently after a certain point in training and (2) to do local updates taking advantage of this frozen variance. This also eliminates the warmup period during which one-bit adam must maintain adam state at full precision.

**Strengths:** The technique is a nice incremental improvement on one-bit adam that further reduces bandwidth requirements, and - in bandwidth-restricted settings - training time. It appears to work quite well, producing similar model performance to one-bit adam.

**Weaknesses:** The paper introduces new hyperparameters that need to be set. This may be annoying for one-off runs, but I suspect that it won't be too hard to find good settings before large hero runs. It would be nice if the paper showed how to efficiently find hyperparameter values here, or even explored whether values found in small-scale models (e.g., 125M GPT model) are applicable to larger-scale models (e.g., 1B GPT model). It would be nice to clarify whether these costs will make the technique difficult to use in practice.

**Suggestions for improving the paper:** The paper could really benefit from having a standard adam baseline as well for context. There's no reason _not_ to have it. The paper could also use error bars and multiple replicates - some of these differences are potentially in the noise. I acknowledge that these are potentially experiments, but I implore the authors to make their best effort here so that the paper is easier for future researchers to build on.

**Note From Pc:**

if the above contains the word "oral" or "spotlight" please see: "oral" presentation means -> notable-top-5% and "spotlight" means -> notable-top-25%. As stated in our emails, we are disassociating presentation type from AC recommendations

**Summary Of Ac-Reviewer Meeting:**

N/A